# Clarify Before You Draw: Proactive Agents for Robust Text-to-CAD Generation

Bo Yuan [1]   Zelin Zhao [1]   Petr Molodyk [1]   Bin Hu [2]   Yongxin Chen [1]

## Abstract

Large language models have recently enabled text-to-CAD systems that synthesize parametric CAD programs (e.g., CadQuery) from natural-language prompts. In practice, however, geometric descriptions can be under-specified or internally inconsistent: critical dimensions may be missing and constraints may conflict. However, existing fine-tuned models tend to reactively follow the user's instructions and hallucinate dimensions when the text is ambiguous. To address this, we propose a proactive agentic framework for text-to-CadQuery generation, named as **Pro-CAD**, that resolves specification issues before code synthesis. Our framework pairs a proactive clarifying agent, which audits the prompt and asks targeted clarification questions only when necessary to produce a self-consistent specification, with a CAD coding agent that translates the specification into an executable CadQuery program. We fine-tune the coding agent based on a curated high-quality text-to-CadQuery dataset and train the clarifying agent via agentic SFT on clarification trajectories. Experiments show that proactive clarification significantly improves robustness to ambiguous prompts while keeping interaction overhead low. ProCAD outperforms frontier closed-source models, including Claude Sonnet 4.5, reducing the mean Chamfer distance by 79.9% and lowering the invalidity ratio from 4.8% to 0.9%. Our code and datasets are made publicly available on `https://github.com/BoYuanVisionary/Pro-CAD`.

## 1. Introduction

Computer-Aided Design (CAD) is central to modern engineering and manufacturing, enabling precise, editable 3D models that support downstream simulation and fabrication (Brière-Côté et al., 2012). Yet creating CAD models remains labor-intensive and expertise-heavy, making rapid iteration costly and limiting accessibility (Robertson & Allen, 2002). Recently, the usage of text-to-CAD methods has gained popularity as they use natural language as an intuitive interface for CAD model creation, potentially lowering the expertise barrier and enabling faster iteration by translating user descriptions into parametric CAD models (Badagabettu et al., 2024; Li et al., 2024a; Khan et al., 2024b). In most existing CAD generation methods, CAD models are represented either as parametric command sequences or as B-rep representations (Wu et al., 2021; Xu et al., 2024; Khan et al., 2024a).

With the advances in large language models (LLMs) and vison language models (VLMs) for language understanding and program synthesis (Chen, 2021; Austin et al., 2021; Jiang et al., 2024), language models have also been applied to text-to-CAD to translate natural language instructions into structured CAD code (Xie & Ju, 2025; Guan et al., 2025; Kolodiazhnyi et al., 2025; Jia et al., 2025). Among various CAD code representations, CadQuery (CadQuery Contributors, 2024), a Python-based parametric scripting language, has been increasingly adopted in recent work (Niu et al., 2025; Kolodiazhnyi et al., 2025; Xie & Ju, 2025; Guan et al., 2025), mostly because modern LLMs tend to be particularly effective at generating Python code (Qing et al., 2025). Recent CadQuery-based methods have also demonstrated strong performance on text-to-CAD generation benchmarks (Kolodiazhnyi et al., 2025; Guan et al., 2025). In this work, we represent CAD models as executable CadQuery programs, following the line of prior work.

The majority of prior works focuses on training LLMs or VLMs to generate CadQuery code, typically assuming that the input text can precisely specify the target shape. However, in practice, natural-language geometric descriptions are often under-specified or internally inconsistent: critical dimensions may be omitted and constraints may conflict, so generation can fail even when the intended shape is simple (Becattini et al., 2013; Cheong et al., 2014). Prior text-to-CAD methods largely follow two paradigms—one-shot generation via fine-tuned LLMs (Zhang et al., 2024; Li et al., 2025a) and feedback-based refinement using parametric or visual signals (Alrashedy et al., 2024; Li et al., 2025b). Both

[1]Georgia Institute of Technology [2]University of Illinois Urbana-Champaign. Correspondence to: Yongxin Chen <yongchen@gatech.edu>.

*Proceedings of the $43^{rd}$ International Conference on Machine Learning*, Seoul, South Korea. PMLR 306, 2026. Copyright 2026 by the author(s).

paradigms typically treat the user prompt as a reliable specification, implicitly requiring users to provide precise and consistent geometric constraints.

This issue motivates a shift from static text-to-CAD generation with a single LLM to dynamic, proactive clarification with an agentic system. Proactive agents not only follow the user's request to improve generation success, but also identify missing or conflicting information and ask only essential clarification questions to minimize interruption and frustration (Lu et al., 2024; Sun et al., 2025). To mitigate the ambiguity in text descriptions, we design a two-agent system consisting of a proactive clarifying agent and a CAD code generation agent. The clarifying agent first audits the user prompt, interacts with the user to resolve missing and conflicting dimensions, and then produces a complete, self-consistent text specification, which is finally passed to the coding agent to generate the CAD program.

To build this proactive agentic system, we fine-tune open-source models on a curated, high-quality text-to-CadQuery dataset of 10K unambiguous samples. We develop a new data creation pipeline that generates human-like, concise natural-language specifications for CAD models from Deep-CAD (Wu et al., 2021). Unlike Text2CAD (Khan et al., 2024b), which is built from a minimal JSON representation, our approach instead uses CadQuery code as the canonical representation. We also apply LLM-based verification and human checks to remove potential data leakage and ensure consistency between the natural language description and the intended geometry. The resulting high-quality text-to-CadQuery dataset is used to fine-tune the CAD coding agent. On top of this, we construct a synthetic dataset to simulate ambiguous text descriptions by perturbing the verified specifications to induce syntactic ambiguity, while keeping the corresponding corrected specifications as targets; this dataset is used to supervise the proactive agent to ask minimal questions and produce a finalized specification.

We summarize our main contributions as follows:

1. We propose a shift from static one-shot generation or post-hoc refinement to a dynamic proactive agentic framework for text-to-CadQuery, where a clarifying agent detects missing or conflicting constraints and asks minimal clarification questions, and a coding agent then synthesizes executable CadQuery code from the resulting specification.

2. We fine-tune our coding agent, *ProCAD-coder*, using only 1.6K carefully curated samples, yet achieve superior performance on unambiguous text-to-CadQuery generation. These 1.6K samples are drawn from a new data-creation pipeline that produces a curated 10K text-to-CadQuery dataset, where specifications are screened with LLM-based checks and human verifications.

3. We fine-tune our clarifying agent, *ProCAD-clarifier*, via agentic SFT on a synthetic dataset of 6,063 samples containing full agentic trajectories to resolve ambiguous prompts. The resulting system outperforms frontier models, including Claude Sonnet 4.5 and GPT-4o-mini, on communication cost, corrected-prompt quality, and downstream geometry quality.

## 2. Related Work

**Text-to-CAD and Parametric Generation.** Early CAD generation relied on static representations like voxels or meshes (Wu et al., 2021), while recent approaches focus on parametric, editable command sequences (Khan et al., 2024a). Text-to-CAD has emerged to lower entry barriers (Badagabettu et al., 2024; Li et al., 2024a; Khan et al., 2024b) and typically treats generation as a direct translation problem. Consequently, current methods can struggle with ambiguous, incomplete, or inconsistent prompts (Becattini et al., 2013).

**LLMs for CAD Generation.** Recent works leverage LLMs to generate structured CAD scripts (Xie & Ju, 2025; Guan et al., 2025; Kolodiazhnyi et al., 2025; Jia et al., 2025), with CadQuery gaining traction due to its Python-based syntax (CadQuery Contributors, 2024; Niu et al., 2025; Qing et al., 2025). Unlike existing one-shot systems (Kolodiazhnyi et al., 2025; Guan et al., 2025) or those relying on post-hoc execution feedback (Alrashedy et al., 2024; Li et al., 2025b; Wang et al., 2025), our work introduces a method to audit and correct specification errors *before* code generation, preventing downstream failures.

**Text-to-CadQuery Datasets.** High-quality datasets pairing expert descriptions with executable CadQuery code are scarce. Existing resources like LLM4CAD (Li et al., 2024b) and Query4CAD (Badagabettu et al., 2024) are often small or limited in scope. Many studies rely on expert-level descriptions in Text2CAD (Xie & Ju, 2025; Guan et al., 2025; Khan et al., 2024b), but these are frequently verbose, noisy, or contain misleading scaling operations (Govindarajan et al., 2025) (see Appendix B). Previous attempts to pair these descriptions with CadQuery code (Kolodiazhnyi et al., 2025; Rukhovich et al., 2025) often overlook critical discrepancies in units and commands. We provide a more detailed discussion of related works in Appendix A.

## 3. Proactive Agentic System

In this work, we study *text-to-CadQuery* generation, where a natural-language specification $p$ is translated into an executable CadQuery program $y$. We allow the specification $p$ to be ambiguous, yet aim to recover the correct CadQuery code with minimal user interruption: if $p$ is fully specified,

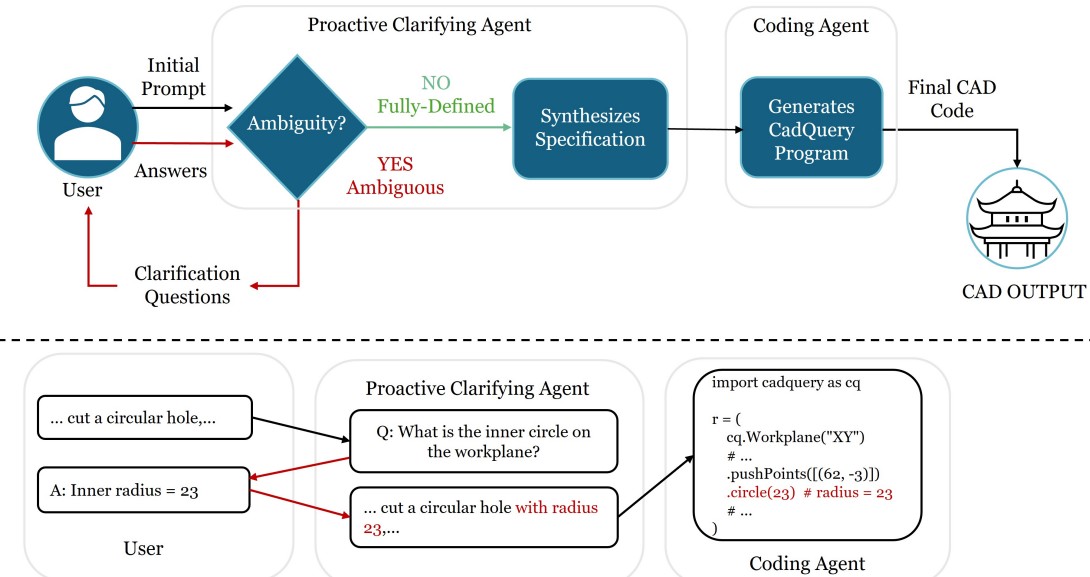

*Figure 1.* Diagram of our two-agent text-to-CadQuery pipeline. A proactive clarifying agent audits the user prompt, asks targeted clarification questions when needed, and outputs a standardized specification, which a coding agent then outputs a CadQuery program. We also provide one example where the ambiguous prompt does not include the radius of the inner circle.

we directly generate $y$; otherwise, we proactively ask for clarification from users before generating $y$.

To this end, instead of relying on a single model to resolve an ambiguous prompt end-to-end, we decompose the task into three explicit stages: (1) detecting ambiguous aspects of the description and asking targeted clarification questions; (2) incorporating the user's feedback to produce a corrected text description; and (3) generating the final CadQuery program from the corrected description. This decomposition makes the process more controllable and interpretable and fits the high standards of engineering design. To implement this paradigm, we propose a two-agent system consisting of a proactive clarification agent and a CAD coding agent, as shown in Figure 1.

More formally, we model the proactive clarification agent $\pi_\phi$ as a finite-horizon Markov decision process $\mathcal{M} = (\mathcal{S}, \mathcal{A}, R)$ where the environment corresponds to the user. We omit the transition kernel for brevity. The interaction starts from the original user prompt $p$. A state $s \in \mathcal{S}$ captures the current context, $s = (p, h)$, where $h$ is the conversation history consisting of previously asked questions and user answers (with $h = \emptyset$ at the start). At each round, the proactive agent $\pi_\phi$ either accepts the current specification or asks a clarification question:

$$a \in \mathcal{A} = \{\text{ACCEPT}\} \cup \{\text{ASK}(u) : u \in \mathcal{U}\}, \quad (1)$$

where $\mathcal{U}$ denotes the space of natural-language questions. If $a = \text{ASK}(u)$, the user provides an answer $v$ and the history is updated as $h \leftarrow h \cup \{(u, v)\}$. When the agent chooses ACCEPT, it outputs a finalized, self-consistent

specification $\hat{p}$ based on $(p, h)$, which is then passed to the coding agent $\pi_\theta$ to generate the CadQuery program $y$.

The two-agent system should maximize the reward that captures both the geometric fidelity of the model and communication overhead with the user. Let $\text{CD}(y)$ denote the Chamfer distance between the generated mesh from code $y$ and the ground-truth mesh, and let $C(h)$ be a nonnegative cost that measures interaction burden, such as the number of rounds, total token length, or latency. We define the reward of the clarifying agent as $R = -\text{CD}(y) - \lambda\, C(h)$, where $\lambda \geq 0$ controls the trade-off between reconstruction accuracy and interaction cost. The objective is to learn policies $\pi_\phi$ and $\pi_\theta$ that maximize the expected return, equivalently minimizing CD while keeping $C(h)$ small. For each step, we carefully design system prompts to enforce format completeness and clearly specify each agent's role. Full prompts are provided in Appendix L.

Our agentic system serves as a flexible framework that can be instantiated with different combinations of a clarifying agent and a CAD coding agent, using either commercial or open-source models. To further improve performance, we design a two-stage training process for both agents using a carefully curated text-to-CadQuery dataset that includes both unambiguous and ambiguous text prompts (Section 4). First, we fine-tune the coding agent on our high-quality text-to-CadQuery dataset of unambiguous text descriptions (Section 5.1). Second, we generate synthetic expert agentic trajectories for resolving ambiguity in natural-language descriptions and use them to fine-tune the clarification agent via agentic supervised fine-tuning (Section 5.2). Our agentic

system, ProCAD, pairs a fine-tuned Qwen2.5-7B-Instruct model (ProCAD-clarifier) as the clarifying agent with another fine-tuned Qwen2.5-7B-Instruct model (ProCAD-coder) as the coding agent, and outperforms even frontier coding models, Claude Sonnet 4.5 (Anthropic, 2025b) in both communication cost and geometric fidelity (Section 6).

## 4. Data Annotation pipeline for high-quality text-to-CadQuery dataset

In contrast to prior pipelines (as discussed in Section A.3) that generate CadQuery code from text descriptions, we instead start from the raw CadQuery programs and generate precise, high-quality text descriptions. CadQuery is a highly interpretable programming language that typically uses common CAD operations to construct geometry. As a result, it contains the complete information needed to reconstruct an accurate and detailed textual specification. We observe that Rukhovich et al. (2025) reconstructs Cad-Query programs for DeepCAD models from point clouds and builds a CadQuery dataset of approximately 17K samples. Building on this, we render the shape from four different viewpoints and prompt a strong vision-language model, GPT-5-mini (Singh et al., 2025), with both the images and the CadQuery program to produce the corresponding text.

We first apply standard deduplication procedures (Xu et al., 2022; 2023) to the original DeepCAD shapes. We only keep the subset of deduplicated shapes for which CadQuery programs are available from (Rukhovich et al., 2025). Finally, we filter out samples whose generated geometry deviates from the reference shape by more than a preset Chamfer distance threshold. This step removes only a small fraction of samples, indicating that the adopted CadQuery corpus is generally of high quality. See Table 7 for details.

At the same time, using CadQuery as an input introduces a new challenge for building text-to-CadQuery data: the generated description may leak code snippets from the source program. To mitigate this risk, our system prompt explicitly instructs the model to produce natural descriptions *without* reproducing any CadQuery surface form. The system prompt is provided in Appendix L.2. In addition, we adopt a generate-then-verify pipeline (Madaan et al., 2023) to detect and filter potential leakage: each generated text will have one data leakage check. This LLM-based check is designed to catch raw code or near-verbatim fragments, while avoiding overly strict false positives—for example, it does *not* flag ordinary geometric terms, e.g., origin, workplane or coordinate tuples as leakage in isolation. Details of the leakage-check prompt are also provided in Appendix L.2.

To further ensure that the generated natural language description is complete and unambiguous, we add an LLM based completeness check. Concretely, we provide the gen-erated description alone to GPT-5-mini without the original CadQuery program and prompt it to synthesize CadQuery code. We then execute the generated program to obtain a three dimensional mesh and compute the Chamfer distance to the ground truth mesh.

Note that this completeness check can be overly strict: even prompts with fully specified descriptions may fail due to the model's limitations in generating correct CadQuery code. Motivated by inference time scaling laws (Wu et al., 2025; Snell et al., 2025), where models often succeed given multiple independent attempts, we incorporate a simple and efficient retry mechanism. If a sample fails either the leakage check or the completeness check, we rerun the entire generation and validation loop up to three times, each time sampling a new description and reapplying both checks.

If all three attempts fail, we escalate this CAD model to a team of CAD experts for manual review and supervision. This design balances the trade off between human effort and automated evaluation during data generation. Empirically, more than 80% of samples pass both checks with retry, which substantially reduces the amount of manual intervention required. See Figure 2 for the complete pipeline.

Compared with the Text2CAD pipeline (Khan et al., 2024b), our approach offers three key advantages. First, we jointly condition the description generator on *both* the multi-view renderings and the CadQuery program, whereas Text2CAD uses visual and symbolic signals separately; this joint conditioning yields descriptions that are better grounded in the underlying geometry and less likely to omit critical constraints (Ngiam et al., 2011). Second, we rely on a frontier VLM, GPT-5-mini, which we find produces substantially more consistent and higher-quality descriptions than smaller models, reducing noise in the resulting supervision. Third, we incorporate two checks for data leakage and completeness with a retry mechanism to further improve quality. To avoid biasing the dataset toward overly simple samples, we additionally route cases that repeatedly fail these checks to human experts.

## 5. ProCAD Training

### 5.1. Coding Agent Training

To train the coding agent, we perform standard supervised fine-tuning (SFT) of an open-source pretrained language model on a paired dataset $\mathcal{D} = \{(p, y)\}$, where $p$ is an unambiguous prompt obtained in Secion 4 and $y$ is the corresponding CadQuery program released by Rukhovich et al. (2025). Let $P_\theta(\cdot \mid p_0, p)$ denote the causal language model distribution over CadQuery programs conditioned on the system prompt $p_0$ and the input prompt $p$. Here $p_0$ specifies the required output format and coding conventions that the generated CadQuery program must follow. We

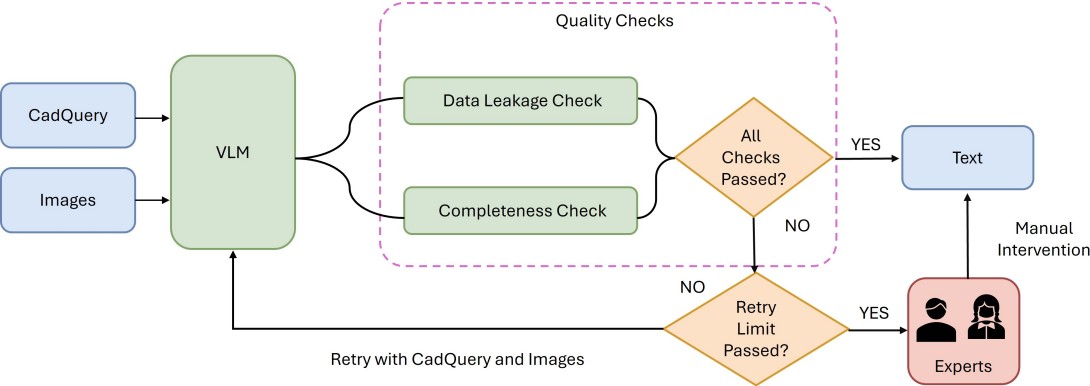

*Figure 2.* Semi-automatic annotation pipeline: render each shape and use its CadQuery to prompt a frontier VLM for a text description; accept only if it passes (i) code-leakage filtering and (ii) completeness verification by regenerating CadQuery with Chamfer distance below a threshold; otherwise retry up to a limit. If the maximum number of retries is reached, we defer the case to CAD experts for manual review.

minimize the negative log-likelihood objective

$$\min_\theta \ \mathcal{L}(\theta) = \mathbb{E}_{(p,y)\sim\mathcal{D}} \left[ -\log \pi_\theta(y \mid p_0, p) \right]. \quad (2)$$

### 5.2. Proactive Clarifying Agent Training

Real user inputs are often noisy and may be under specified or contain intrinsically contradictory dimensions. Training a coding agent with standard SFT alone is therefore insufficient to reliably detect and resolve such specification errors. A natural mitigation is to incorporate richer feedback (Al-rashedy et al., 2024; Li et al., 2025b; An et al., 2026), such as images or point clouds of the rendered models as targets, but these signals are often too coarse to enforce precise metric constraints and may not penalize small yet consequential deviations. We argue that this mismatch is especially critical in CAD design, where even minor dimensional errors can violate product requirements and cause downstream manufacturing failures.

In practice, fully specifying every dimension of a complex CAD part can be difficult for users, whereas providing a few dimensions in response to targeted questions is often easier. Hence in our experiments, we assume that the user can provide correct answers to any asked question as long as the question itself is clear. Under this assumption, an optimal policy should minimize the number of interaction rounds, since additional rounds only increase communication cost and lengthen the context presented to the agent. Consequently, the general multi-round optimization reduces to a two-round policy: in the first round, the agent either directly accepts the original prompt and output the corrected text $\hat{p} = p$, or asks a set of targeted clarification questions $\{u_j\}$ in a single message. The question should be clear and specific enough that users can give right dimensions. In the second round, after receiving the user answers, the agent deterministically accepts and outputs the corrected specification $\hat{p}$, which is then passed to the coding agent.

We train the clarifying agent from two kinds of supervision, and in both cases the expert target is a JSON-formatted output. For unambiguous prompts, the dataset is $\mathcal{D}_{\mathrm{acc}} = \{p^{(i)}, y_{\mathrm{acc}}^{(i)}\}$, where the target JSON is $y_{\mathrm{acc}}^{(i)} = \{\text{is misleading} : \text{False, standardized prompt} : p^{(i)}\}$. For ambiguous prompts, we store clarification trajectories in the form $(\hat{p}^{(j)}, \mathbf{q}^{(j)}, \mathbf{a}^{(j)}, p^{(j)})$, where $\mathbf{q}^{(j)}$ are the clarification questions and $\mathbf{a}^{(j)}$ are the corresponding user answers. Supervision is provided in two JSON outputs. The first JSON supervises question generation, $y_{\mathrm{ask}}^{(j)} = \{\text{is misleading} : \text{True, questions} : \mathbf{q}^{(j)}\}$, and the second JSON supervises the final corrected specification, $y_{\mathrm{clr}}^{(j)} = \{\text{is misleading} : \text{True, standardized prompt} : p^{(j)}\}$.

We train the model $\pi_\phi$ to reproduce these JSON outputs via maximum likelihood. The overall objective sums the three losses where $s_\phi$ is the system prompt, as shown in Appendix L.1.

$$\mathcal{L}(\phi) = \mathbb{E}_{(p,y_{\mathrm{acc}})\sim\mathcal{D}_{\mathrm{acc}}} \left[ -\log \pi_\phi \big( y_{\mathrm{acc}} \mid s_\phi, p \big) \right]$$
$$+ \mathbb{E}_{(\hat{p},y_{\mathrm{ask}})\sim\mathcal{D}_{\mathrm{clr}}} \left[ -\log \pi_\phi \big( y_{\mathrm{ask}} \mid s_\phi, \hat{p} \big) \right]$$
$$+ \mathbb{E}_{(\hat{p},\mathbf{q},\mathbf{a},y_{\mathrm{clr}})\sim\mathcal{D}_{\mathrm{clr}}} \left[ -\log \pi_\phi \big( y_{\mathrm{clr}} \mid s_\phi, \hat{p}, \mathbf{q}, \mathbf{a} \big) \right]. \quad (3)$$

## 6. Experiments

**Metrics** In our experiments, we report two primary metrics. **Chamfer distance** (CD) measures geometric fidelity between the generated and reference shapes. **Invalidity Ratio** (IR) is the percentage of generated samples that cannot be executed or rendered into valid CAD objects. Both follow common practice in previous works (Guan et al., 2025; Kolodiazhnyi et al., 2025; Xie & Ju, 2025; Wang et al., 2025). In addition, we use GPT-5-mini as a judge to assess the quality of our text-to-CadQuery dataset and the effectiveness of the resulting user interactions.

## 6.1. Text-to-CadQuery dataset

**Dataset Creation**  To construct our high-quality Text-to-CadQuery dataset, we start from the CadQuery dataset of Rukhovich et al. (2025) and retain only samples whose reconstructed geometry matches the reference with CD below $2 \times 10^{-4}$. The full CD distribution is reported in Table 7. We use the same threshold in the completeness check in the data annotation pipeline, where we require that, given only the generated natural-language description, GPT-5-mini can synthesize CadQuery code whose executed geometry attains CD $< 2 \times 10^{-4}$. For each sample, we allow up to three retries in the generation-and-verification loop, and find that over $80\%$ of samples pass both the leakage and completeness checks without human intervention. The system prompts used in our annotation pipeline are provided in Appendix L.2. Each generated description follows a fixed structure with three parts: General shape, Setup, and Build description. General shape briefly names the part and its main features, Setup specifies the workplane and its origin including any coordinate transforms, and Build description provides step-by-step instructions.

**Comparison against Text2CAD**  Table 3 compares our dataset with Text2CAD. First, our prompts are substantially shorter, primarily because we retain only the information necessary to reconstruct the CadQuery program, whereas Text2CAD descriptions are designed for command-sequence generation and often include redundant details. Second, we use an LLM-as-judge to assess *clarity* and *human-likeness*. We randomly sample 1,000 pairs and ask the judge to choose which description is better under each criterion. Because LLM judges can exhibit position bias (Zheng et al., 2023; Shi et al., 2025), where preferences depend on whether an option appears first or second, we evaluate both presentation orders and report results for "Ours first" and "Text2CAD first." Across both orders, our prompts are consistently preferred in terms of human-likeness. For clarity, Text2CAD descriptions typically include redundant details, e.g., Euler angles, scaling factors, and translation vectors, that can distract from the core geometric specification, whereas our prompts are more concise and less confusing, leading to higher clarity win rates as well. Finally, it is worth noting that LLM-based judges often exhibit a length bias, tending to prefer longer responses (Saito et al., 2023; Dubois et al., 2024). Since our prompts are markedly shorter, these preference-based evaluations may be biased against our dataset, yet we still outperform Text2CAD, underscoring our data quality. Prompts for LLM-as-judge are shown in Appendix L.4.

**Zero-shot performance**  Moreover, we evaluate the zero-shot performance of both frontier models and open-source models on Text2CAD and our new dataset, as in Table 1.

*Table 1.* Invalidity ratio (IR) and Chamfer distance (CD) on Ours and Text2CAD. CD is reported in units of $\times 10^3$ (lower is better).

| Model | Dataset | IR (%)↓ | Mean CD↓ | Median CD↓ |
|---|---|---|---|---|
| Claude 4.5 Sonnet | Ours | **11.8** | **2.580** | **0.074** |
| | Text2CAD | 56.6 | 27.244 | 12.549 |
| Qwen2.5 7B Instruct | Ours | 82.9 | 4.489 | **0.100** |
| | Text2CAD | 86.9 | **4.434** | 0.103 |

For a fair comparison, we consider the subset of Text2CAD that shares the same shape UIDs as our data. We find that Qwen2.5-7B-Instruct (Bai et al., 2025) performs poorly in zero-shot settings on both datasets, with an invalidity ratio of nearly $85\%$. In contrast, Claude Sonnet 4.5 (Anthropic, 2025b), a strong frontier coding model, achieves substantially better results: the zero-shot invalidity ratio decreases from approximately $54.5\%$ on Text2CAD to about $13\%$ on our dataset, and it also attains much lower mean and median Chamfer distances. These results partially support our claim that our dataset is easier to follow and contains fewer misleading specifications. We also observe that most failures of Qwen2.5-7B-Instruct in the zero-shot setting are due to CadQuery syntax errors rather than geometric reasoning. In the following experiments we use Qwen2.5-7B-Instruct as the base model and fine-tune it on our dataset to improve code validity and overall generation quality.

### 6.2. ProCAD-coder

**Setup**  We sample 1.6K examples for training and 1K examples for testing from our 10K dataset. Our coding agent is initialized from Qwen2.5-7B-Instruct, which takes a natural-language prompt as input and outputs a standardized CadQuery program. We perform full-parameter fine-tuning on two H200 GPUs with batch size 16, learning rate $10^{-5}$, and two training epochs. This results in 200 total optimization steps and completes in under 10 minutes.

We find that even this lightweight fine-tuning already yields large gains on the test set: the invalidity ratio drops from nearly $86.9\%$ to $0.9\%$, while the median Chamfer distance reaches $6.6 \times 10^{-5}$. Notably, this result is competitive with, and in some cases better than, Claude 4.5 Sonnet, which attains an invalidity ratio of about $13\%$ with median Chamfer distance $7.7 \times 10^{-5}$. By comparison, prior work typically relies on more than 100K supervised examples, often combined with additional refinement or reinforcement learning, to achieve similar reliability (Xie & Ju, 2025; Guan et al., 2025; An et al., 2026; Kolodiazhnyi et al., 2025). We attribute our strong performance to both our high-quality data creation pipeline and the use of a strong base model. Notably, even when using the same Qwen2.5 backbone for a fair comparison, prior work trains on more than 150K samples (Guan et al., 2025) and additionally applies both SFT and reinforcement learning. See Appendix F for compre-

*Table 2.* Performance on the 1K unambiguous prompts. CD is reported in units of $\times 10^3$ (lower is better).

| Method | IR (%)↓ | Mean CD↓ | Median CD↓ |
|---|---|---|---|
| Ours | **0.9** | **0.108** | **0.066** |
| Text-CAD | 14.5 | 3.054 | 0.097 |
| JSON-Distill | 5.3 | 23.117 | 8.808 |
| Claude 4.5 Sonnet | 12.9 | 1.580 | 0.077 |

*Table 3.* Prompt length statistics and LLM-judge preference win rates (%) for Ours vs. Text2CAD.

| Metric | Ours | Text2CAD |
|---|---|---|
| **Length statistics** ↓ | | |
| Mean length | **147.8** | 285.4 |
| Median length | **119.0** | 228.0 |
| **Win rate %**, ↑ | | |
| Clarity, Ours first | **98.4** | 1.6 |
| Clarity, Text2CAD first | **66.0** | 34.0 |
| Human-likeness, Ours first | **100.0** | 0.0 |
| Human-likeness, Text2CAD first | **96.5** | 3.5 |

hensive comparison.

To ensure a fair comparison, we keep the underlying CAD shapes fixed and use the same training and test splits and identical fine-tuning hyperparameters; we vary only the data construction pipeline, which yields different text descriptions and CadQuery program representations for the same shapes. We consider two baselines: **Text-CAD**, which follows Kolodiazhnyi et al. (2025) by directly pairing the expert-level Text2CAD prompts with the CadQuery programs from Rukhovich et al. (2025); and **JSON-Distill**, which uses the open-source dataset of Xie & Ju (2025) where the text is the expert-level Text2CAD description and the CadQuery program is distilled from Gemini 2.0 Flash based on the minimal JSON representation.

As shown in Table 2, our experimental results further demonstrate the importance of text quality in the text-to-CadQuery task. Moreover, our model outperforms Claude Sonnet 4.5 across all evaluation metrics. While most prior work focuses on improving the model and training procedure on Text2CAD with various techniques, our findings suggest that a key bottleneck also lies in the quality of the original text descriptions. In particular, the code generation model can be strong enough to produce valid CadQuery programs when the input specification is clear and correct. This observation motivates us to consider a more realistic setting in which user prompts may be ambiguous, and to introduce a proactive clarifying agent that detects specification issues and asks targeted questions before code generation.

### 6.3. ProCAD-clarifier

After training ProCAD-coder on our new Text2CadQuery dataset, we build the full agentic system. Specifically, to build ProCAD-clarifier, we fine-tune Qwen2.5-7B-Instruct using agentic SFT, as described in Section 5.2.

Modeling real user behavior for proactive clarification is extremely challenging: it typically requires large-scale interactive data collection and can exhibit substantial variability across annotators (Testoni & Fernández, 2024; Ito et al., 2025; Sahay et al., 2025). As the first work toward studying clarification for ambiguous CAD prompts, we therefore adopt a scalable alternative and use GPT-5-mini as a user simulator, following prior work that leverages LLM-based user simulation in dialogue systems (Sekulić et al., 2024) and recommendation settings (Zhang et al., 2025). This enables us to synthesize ambiguous prompts with predefined ambiguity types in a controllable and reproducible way, while avoiding the cost and noise of large-scale human interaction data.

To generate ambiguous prompts, we prompt GPT-5-mini with a detailed system instruction to perturb the original, verified specifications from our Text-to-CadQuery dataset. See system prompts in Appendix L.3. We focus on two common error types in CAD practice: (i) *under-specified* prompts that omit key dimensions, and (ii) *inconsistent* prompts that assign conflicting values to the same feature.

For each new prompt, we send the it back to GPT-5-mini for a self-refine (Madaan et al., 2023) to filter obvious errors and improve consistency. We then curate a subset of representative cases using three selection rules. Specifically, for each ambiguous prompt $\hat{p}$, we run ProCAD-coder to synthesize a CadQuery program and compute its CD against the ground-truth mesh. We keep a sample only if: (i) the original verified specification $p$ is high-quality, with $CD < 2 \times 10^{-4}$; (ii) the perturbed prompt $\hat{p}$ is genuinely harmful, with $CD > 2 \times 10^{-4}$; and (iii) the degradation is substantial: the ratio of the two Chamfer distances is at least 10. In addition, we include unambiguous prompts to balance the dataset. Using this pipeline, we construct a training set of 6,063 samples and a test set of 2,469 samples, with an approximately 1:1 ratio between unambiguous and ambiguous prompts. See Table 8 for detailed statistics.

We train ProCAD-clarifier using the same base model and fine-tuning hyperparameters as ProCAD-coder. With batch size 16, training takes 367 optimization steps and finishes in under 10 minutes. Table 4 reports the performance of our agentic system on the 2,469 test samples. We compare against two classes of baselines: (i) single-model, and (ii) agentic baselines . For all agentic variants, we keep the coding agent fixed as either our fine-tuned ProCAD-coder or Claude 4.5 Sonnet, since they achieve strong performance

*Table 4.* Performance on the test set with 2,469 ambiguous prompts, where user responses are simulated by GPT-5-mini. In the two-agent setting (Figure 1), we fix the coding agent and vary the clarification agent. Bold and underline denote the best and second-best values.

| Setting | Model | Efficiency ↑ | Resolution ↑ | Mean CD ↓ | Median CD ↓ | IR % ↓ |
|---|---|---|---|---|---|---|
| **Single-model** | Cadrille (Kolodiazhnyi et al., 2025) | – | – | 55.43 | 44.92 | 20.7% |
| | Qwen 2.5-7B-Instruct | – | – | 10.94 | 1.03 | 68.2% |
| | GPT-4o-mini | – | – | 12.58 | 0.78 | 28.7% |
| | Claude Sonnet 4.5 | – | – | 7.80 | 0.19 | 14.6% |
| **Two-agent** (coding=Claude 4.5 Sonnet) | Qwen 2.5-7B-Instruct | 0.6606 | 0.6487 | 11.56 | 0.33 | 4.5% |
| | GPT-4o-mini | 0.5788 | 0.7814 | 9.98 | 0.14 | 7.5% |
| | Claude Sonnet 4.5 | 0.8255 | 0.9329 | 3.10 | 0.09 | 4.8% |
| | **ProCAD-clarifier (Ours)** | **0.9665** | 0.9327 | 0.85 | **0.08** | 4.0% |
| **Two-agent** (coding=ProCAD-coder) | Qwen 2.5-7B-Instruct | 0.6706 | 0.6597 | 11.68 | 0.33 | 4.1% |
| | GPT-4o-mini | 0.5677 | 0.7712 | 9.38 | 0.12 | 3.3% |
| | Claude Sonnet 4.5 | 0.8485 | 0.9120 | 2.69 | **0.08** | 2.3% |
| | **ProCAD-clarifier (Ours)** | 0.9654 | **0.9341** | **0.63** | **0.08** | **0.9%** |

*Table 5.* Performance on the test set with ambiguous prompts, where user responses are simulated by Claude 4.5 Haiku for out-of-distribution task. In the two-agent setting, we fix the coding agent and vary the clarification agent.

| Setting | Model | Efficiency ↑ | Resolution ↑ | Mean CD ↓ | Median CD ↓ | IR % ↓ |
|---|---|---|---|---|---|---|
| **Two-agent** (coding = Claude 4.5 Sonnet) | Claude Sonnet 4.5 | 0.8249 | 0.9367 | 3.06 | 0.09 | 4.2% |
| | **ProCAD-clarifier (Ours)** | **0.9668** | 0.9354 | 0.63 | **0.07** | 4.0% |
| **Two-agent** (coding = ProCAD-coder) | Claude Sonnet 4.5 | 0.8298 | 0.9372 | 3.14 | 0.08 | 1.7% |
| | **ProCAD-clarifier (Ours)** | 0.9658 | **0.9415** | **0.46** | **0.07** | **0.9%** |

*Table 6.* Human-centered evaluations. Efficiency and Resolution are scored by human experts. CD is reported in units of $\times 10^3$ (lower is better). Bold and underline denote the best and second-best values.

| Setting | Model | Efficiency ↑ | Resolution ↑ | Mean CD ↓ | Median CD ↓ | IR % ↓ |
|---|---|---|---|---|---|---|
| **Two-agent** (coding = ProCAD-coder) | Qwen 2.5-7B-Instruct | 0.016 | 0.000 | 13.98 | 3.69 | 14.6% |
| | GPT-4o-mini | 0.251 | 0.287 | 13.81 | 0.14 | 19.5% |
| | Claude Sonnet 4.5 | 0.598 | 0.700 | 9.72 | 0.11 | **12.2%** |
| | **ProCAD-clarifier (Ours)** | **0.760** | **0.787** | **1.28** | **0.09** | **12.2%** |

on unambiguous descriptions, as shown in Table 2.

Beyond Chamfer distance and invalidity ratio, we also evaluate the interaction quality using an LLM-as-judge for the two-agent system. Specifically, we compute an efficiency score that measures whether the clarifier's questions match the ground-truth questions without introducing redundant queries, and a resolution score that measures whether the clarified specification successfully resolves the ambiguity. See Appendix G for more details. We include Cadrille (Kolodiazhnyi et al., 2025) only for completeness, noting it was trained on standard Text2CAD data and lacks ambiguity detection capabilities. As this is the first work to explicitly address ambiguity in text-to-CAD, our primary baselines are general-purpose LLMs.

**Our two-agent system outperforms single-model systems** Table 4 shows that introducing a clarification agent substantially improves robustness to ambiguous inputs. Compared to single-model direct coding, the two-agent pipeline consistently reduces the invalid rate and improves geometric

fidelity, demonstrating the benefit of resolving underspecification and inconsistencies before code synthesis.

**ProCAD achieves the best overall results.** Among all variants, pairing ProCAD-clarifier with ProCAD-coder yields the strongest performance across all metrics: it attains the lowest mean CD and invalid rate while also achieving the highest Efficiency and Resolution scores. Beyond geometric quality, ProCAD-clarifier minimizes user intervention by asking only the most necessary, targeted questions, and it produces higher-quality corrected prompts, which in turn enables more reliable downstream code generation. See Appendix H for case studies where we fix the coding agent as ProCAD-coder and compare Claude Sonnet 4.5 and ProCAD-clarifier as the clarifying agent. See Appendix I for qualitative comparison.

**Generalization to out-of-distribution simulators.** Moreover, we observe that ProCAD exhibits strong generalization capabilities, performing robustly even in out-of-distribution settings. While our training data consists entirely of user

responses simulated by GPT-5-mini, we demonstrate in Table 5 that performance remains high when the user simulator is switched to Claude 4.5 Haiku (Anthropic, 2025a). Pro-CAD consistently outperforms baselines where one of the agents is replaced by Claude 4.5 Sonnet.

### 6.4. Human-Centered Evaluation

The experiments in Tables 4 and 5 rely on LLM-based user simulators for both the ambiguous prompts and the user responses. To verify that the conclusions transfer to real users, we additionally conduct a human-centered evaluation in which both the ambiguous prompts and the user responses are provided by real annotators rather than LLM simulators.

**Setup.** We sample 100 examples from our high-quality Text-to-CadQuery dataset. For each example, we provide the original unambiguous prompt together with the reference image to human annotators, since the original prompt contains the precise dimensions. We then ask multiple annotators to rewrite ambiguous prompts. The resulting clarification questions and corrected prompts are further evaluated by CAD experts. In this human evaluation, we compare ProCAD-clarifier against three baselines while fixing the downstream generation model as ProCAD-coder for all methods.

**ProCAD-clarifier remains the strongest under real human interaction.** As shown in Table 6, ProCAD-clarifier continues to achieve the strongest performance across all metrics in this fully human-driven evaluation. Crucially, the interaction-quality metrics (Efficiency and Resolution) are now scored by human experts rather than an LLM judge, yet the conclusions are unchanged: our method consistently outperforms all three baselines. Since Resolution is closely tied to final geometric fidelity, the improvement in Chamfer Distance provides an additional proxy showing that the clarified prompts produced by ProCAD-clarifier better match the target shape.

**Reliability of LLM-based evaluation.** We also note that our LLM-based evaluation is intentionally designed to reduce judging difficulty. For Efficiency, the judge is asked to compare the clarification questions against the ground-truth missing information rather than make a fully open-ended quality judgment. For Resolution, we use a coarse three-level score $\{0, 0.5, 1\}$, which further reduces sensitivity. To directly quantify reliability, we randomly sample 100 test examples from the synthetic dataset and compare LLM-based evaluation against human evaluation of interaction quality. The two agree on more than $96\%$ of the samples, providing additional evidence that the LLM-based metrics are reasonably reliable in our setting.

### 7. Conclusion

We introduced ProCAD, a novel two-agent framework that proactively addresses ambiguity in text-to-CAD generation. By fine-tuning our agents on a curated dataset of 10k high-quality samples, we demonstrated that resolving specification errors before code generation significantly improves reliability. Our text-to-CadQuery dataset suggests that description quality strongly affects downstream code synthesis: even with the same underlying geometry distribution, cleaner, more precise, and constraint-complete descriptions lead to markedly more reliable CAD programs. This highlights the need for expert-level, human-annotated datasets that reflect real engineering specifications in the future works. Our findings also underscore the necessity of moving beyond static prompting toward dynamic, interactive agents. Future directions include gathering large-scale ambiguity datasets from real-world human interactions and developing dedicated user-simulator models.

### Acknowledgments

We thank the anonymous reviewers for their valuable feedback and constructive suggestions, which helped improve the quality and clarity of this paper.

### Impact Statement

This paper presents work whose goal is to advance the field of Machine Learning. There are many potential societal consequences of our work, none which we feel must be specifically highlighted here.

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

**Appendix Summary**   This appendix provides supplementary details and qualitative analysis to support the main findings. We first show the extended related works in Appendix A. We then analyze specific failure modes associated with explicit scaling operations in Appendix B and present a qualitative case study comparing our natural language descriptions against the Text2CAD baseline in Appendix C. We then report detailed statistics on the ground-truth CadQuery code quality (Appendix D) and the distribution of the ambiguous prompt dataset (Appendix E). Furthermore, Appendix F compares our experimental setup with related works, while Appendix G defines the exact LLM-as-judge metrics used to evaluate the clarification agent. Appendix H lists examples where Our ProCAD outperforms Claude Sonnet 4.5 on resolving. Appendix I shows qualitative comparison against baselines. Then Appendix J shows the failure mode of our system. Appendix K shows that our protocol balances clarification quality and latency best. Finally, Appendix L provides the full set of system prompts used for inference, data annotation, and ambiguity generation.

# A. Related Works

## A.1. Text-to-CAD and Parametric Shape Generation

Early learning-based CAD generation methods focused on synthesizing shapes from structured representations such as voxel grids, meshes, or boundary representations, often without explicit programmatic edit-ability (Wu et al., 2021; Zhao et al., 2024). More recent work has emphasized generating parametric CAD models represented as command sequences or sketch-extrude programs, enabling downstream modification and reuse (Khan et al., 2024a). These approaches typically assume access to clean, fully specified input signals, such as aligned sketches or reference shapes (Khan et al., 2024a).

With the rise of natural-language interfaces, text-to-CAD has emerged as a promising direction for lowering the barrier to CAD modeling (Badagabettu et al., 2024; Li et al., 2024a; Khan et al., 2024b). Existing methods generally cast text-to-CAD as a conditional generation problem, mapping user prompts directly to CAD programs or intermediate representations via supervised learning. While these methods demonstrate impressive results under curated benchmarks, they often struggle when prompts are ambiguous, incomplete, or internally inconsistent—a common case in real-world human descriptions (Becattini et al., 2013).

## A.2. LLMs for CAD Generation

Recent advances in large language models (LLMs) for program synthesis (Zhao et al., 2021a;b; Chen, 2021; Austin et al., 2021; Jiang et al., 2024) have motivated their application to CAD code generation. Several works leverage LLMs to translate natural language into structured CAD scripts, including OpenSCAD, CadQuery, or proprietary CAD-like languages (Xie & Ju, 2025; Guan et al., 2025; Kolodiazhnyi et al., 2025; Jia et al., 2025). Among these, CadQuery has gained particular traction due to its Python-based syntax and compositional structure, which aligns well with LLMs' strong performance on Python code generation (CadQuery Contributors, 2024; Niu et al., 2025; Qing et al., 2025).

Most existing text-to-CadQuery systems follow a one-shot generation paradigm, either using prompt-engineered frontier models or fine-tuned open-source LLMs (Kolodiazhnyi et al., 2025; Guan et al., 2025). Some works incorporate execution feedback or geometric validation to iteratively refine generated code (Alrashedy et al., 2024; Li et al., 2025b). However, these approaches still treat the original user prompt as a fixed specification and rely on post-hoc correction when failures occur. In contrast, our work addresses specification errors *before* code generation by explicitly auditing and completing the textual description, thereby reducing downstream failure modes.

## A.3. Text-to-CadQuery Dataset

To the best of our knowledge, only a small number of prior works provide *both* expert-level natural-language descriptions and ground-truth CadQuery programs. While some datasets include general text descriptions without precise dimensions (Xu et al., 2024; Khan et al., 2024b), producing executable CadQuery code typically requires expert-level, highly precise specifications, making large-scale annotation costly and difficult to scale.

LLM4CAD (Li et al., 2024b) contains roughly 5,000 annotated samples, but it focuses on only five common mechanical part categories. Query4CAD (Badagabettu et al., 2024) is substantially smaller, with just 57 samples. Consequently, many text-to-CadQuery studies build their own annotation pipelines by prompting an LLM or VLM and filtering low-quality programs(Xie & Ju, 2025; Guan et al., 2025), primarily using the expert-level procedural instructions in Text2CAD (Khan et al., 2024b) as the primary text source. However, translating Text2CAD's expert descriptions into CadQuery code is

nontrivial: these descriptions are frequently noisy and overly long, containing redundant details that can distract the model and increase the risk of hallucinated or incorrect code (Govindarajan et al., 2025). In particular, scaling operations frequently result in misleading descriptions, as documented in the failure mode analysis in Appendix B. Although Kolodiazhnyi et al. (2025) pair expert descriptions directly with the CadQuery code from Rukhovich et al. (2025), their approach overlooks the discrepancy between the units and commands specified in the text versus the actual code: in Text2CAD, key dimensions and units are derived from the minimal JSON specification, whereas the CadQuery programs are reconstructed from command sequences. This discrepancy makes it difficult for LLMs to align text and code reliably, especially in a zero-shot setting.

### A.4. Proactive and Agentic Language Models

Proactive agents extend reactive instruction-following models by anticipating user needs, identifying missing or inconsistent information, and initiating clarification before acting (Lu et al., 2024; Sun et al., 2025). Such agentic behaviors have shown benefits in task-oriented dialogue, decision support, and program synthesis, where agents may decompose tasks, validate intermediate results, or iteratively refine specifications. However, their application to geometric modeling and CAD remains limited. We bring proactive agent design into text-to-CAD generation by introducing a dedicated specification agent that audits prompts for completeness and consistency and interacts with the user only when necessary. Trained with domain-specific supervision derived from systematically perturbed CAD specifications, our agent balances robustness to ambiguous prompts with low interaction overhead.

## B. Failure modes of scaling operations in Text2CAD

---

**Original prompt (excerpt):**

*... create a circle with center (**0.1293, 0.1293**) and radius **0.1293**. Then add another concentric circle with radius **0.0853**. After completing the sketch, **apply a scaling factor of 0.2586 to the entire sketch**. ... **extrude** the sketch by **0.75** units along the normal direction. The dimensions of the resulting object are **0.2586** in length, **0.2586** in width, and **0.75** in height.*

---

**Incorrect CadQuery code:**

```python
import cadquery as cq

wp = (cq.Workplane("XY")
      .center(0.1293, 0.1293)
      .circle(0.1293)
      .circle(0.0853))

wp = wp.scale(0.2586)  # incorrect: Workplane has no scale() API
result = wp.extrude(0.75)
```

---

*Figure 3.* One failure example in Text2CAD for text-to-CadQuery generation.

We observe that explicit scaling operations appear in a large fraction of Text2CAD examples. This design choice is historically motivated by earlier sequence-based CAD generation settings, where Transformer models were assumed to represent continuous parameters within a fixed numeric range; consequently, real-valued dimensions were rescaled into a predefined interval to ease tokenization and command-sequence prediction. However, this convention transfers poorly to CadQuery, where geometry is expressed as executable Python code. In CadQuery, scaling is not a generic operation that can be freely applied at any stage: in particular, a naive interpretation such as "scale the workplane" may prompt weaker code generators to hallucinate unsupported APIs, e.g., `Workplane.scale(...)`, yielding invalid programs. We further note that the *ordering* in these prompts makes mistakes particularly likely: the scaling step is described immediately after the 2D sketch construction, rather than after the 3D solid is created via extrusion . This placement encourages an implementation that attempts to scale the workplane. Therefore, the prompt structure itself can systematically bias code generators toward

an invalid code, even when the underlying geometry is simple.

Moreover, as illustrated in Figure 3, the scaling statement admits two plausible but conflicting interpretations:

- **Interpretation A (literal post-sketch scaling).** One may follow the prompt literally by first drawing the circles with center $(0.1293, 0.1293)$ and radii $0.1293$ and $0.0853$, and then scaling the entire sketch by $0.2586$. Under this interpretation, the outer radius becomes $0.2586 \cdot 0.1293 \approx 0.0334$, implying a footprint of approximately $2 \times 0.2586 \cdot 0.1293 \approx 0.0669$. This contradicts the stated final dimensions $0.2586 \times 0.2586 \times 0.75$.

- **Interpretation B (parameters already in the target units).** Alternatively, one may treat the listed coordinates and radii as already expressed in the final unit system, in which case the explicit scaling step is redundant and should be ignored.

## C. Case Study: Comparing Text2CAD with our descriptions

---

**Text Descriptions for a Simple Rectangular Prism**

**Text2CAD**
Create a new coordinate system by setting the Euler angles to negative ninety degrees, zero degrees, and negative ninety degrees, and set the translation vector to zero, zero, and zero. On the first face, draw the first loop consting of four straight lines. The first line starts at the origin (zero, zero) and ends at (zero point 0.75, zero). The second line starts at (zero 0.75, zero) and ends at (zero point, zero 0.5566225). The third line starts at (zero 0.75, 0.5566225) and ends at (zero, zero 0.556625). The fourth line starts at (zero, zero, 0.556225) and returns to the origin. Scale the entire sketch by a factor of zero point .75. Rotate the scaled sketch using the same Euler angles as the coordinate system and translate it using the same translation vector. Extrude the transformed sketch sketch by zero point 0.2281 units along the normal direction to create a new solid body. The final dimensions of the rectangular prism with rounded edges and a slightly tilted top surface are: length zero 0.75 units, width zero 0.556225 units, and height zero 0.02801 125 units.

---

**Ours**
This is a single solid rectangular plate: a plain 200 by 150 face extruded to a thickness of 7 (no holes or additional features).
Setup: workplane is the YZ plane with its origin translated to (-4, -100, -75).
Build: on that YZ workplane sketch a rectangle 200 (along the workplane's first axis) by 150 (along the workplane's second axis) with its lower-left corner at the workplane origin (sketch points at (0,0) to (200,150)). Extrude the rectangle 7 in the positive normal direction to form the solid plate.

---

**CadQuery Code**

```
import cadquery as cq

w0 = cq.Workplane('YZ', origin=(-4, -100, -75))

r = w0.sketch().face(
    w0.sketch()
      .segment((0,0), (200,0))
      .segment((200,0), (200,150))
      .segment((200,150), (0,150))
      .segment((0,150), (0,0))
      .assemble()
).finalize().extrude(7)
```

---

We notice that even for a simple rectangular prism, the Text2CAD description is overly long and unnatural, whereas our generated prompt is more concise and precisely captures the shape specification. Among the three representations (expert-level text in Text2CAD, CadQuery code, and our text), Text2CAD is verbose, CadQuery is concise but abstract, and our representation is both human-like and compact while remaining easy to interpret.

## D. Data Statistics of CadQuery code

We report Chamfer Distance statistics for the CadQuery programs released by Rukhovich et al. (2025). As shown in Table 7, the vast majority of samples reconstruct the target geometry with high fidelity, indicating that these programs can serve as reliable ground-truth code. This provides a practical alternative to relying on the minimal structured JSON representation used in Text2CAD. In particular, over $93\%$ of samples achieve a Chamfer Distance below $2 \times 10^{-4}$.

*Table 7.* Chamfer distance distribution for CadQuery reconstructions from Rukhovich et al. (2025). Percentage denotes the fraction of samples whose Chamfer distance is below the specified threshold.

| CD ($\times 10^3$) | Percentage |
|---|---|
| 0.1 | 69.81% |
| 0.2 | 93.37% |
| 0.5 | 97.95% |
| 1 | 98.63% |
| 2 | 99.07% |

## E. Data Statistics of Ambiguous Prompts

*Table 8.* Train and test split statistics for different ambiguity types. Here, the number of issues refers to the number of dimensions that contain ambiguities.

| | Train (N=6,063) | Test (N=2,469) |
|---|---|---|
| Unambiguous | 3,200 | 1,000 |
| Under-specified (1 issue) | 1,071 | 427 |
| Under-specified (2 issues) | 479 | 638 |
| **total** | **1,550** | **1,065** |
| Conflicting (1 issue) | 989 | 314 |
| Conflicting (2 issues) | 324 | 90 |
| **total** | **1,313** | **404** |

## F. Setup Comparison with existing Text-to-CadQuery works

*Table 9.* Training setup and data sources for Text-to-CadQuery.

| Model | Train # | Base model | Training | Text source | CadQuery source |
|---|---|---|---|---|---|
| ProCAD-coder (ours) | 1.6K | Qwen2.5-7B-Instruct | SFT | GPT-5-mini | CAD-recode |
| Cadrille (Kolodiazhnyi et al., 2025) | 160K | Qwen2-VL-2B | SFT+RL | Text2CAD | CAD-recode |
| PR-CAD (An et al., 2026) | 150K | Qwen2.5-7B-Instruct | SFT+RL | Qwen2.5-72B | Gemini-2.5-Flash |
| Text2CadQuery (Xie & Ju, 2025) | 150K | Qwen2.5-3B | SFT | Text2CAD | Gemini-2.0-Flash |
| CAD-coder (Guan et al., 2025) | 150K | Qwen2.5-7B-Instruct | SFT+RL | Text2CAD | DeepSeek-V3 (Liu et al., 2024) |

*Table 10.* Performance comparison. CD values are scaled by $10^3$; lower is better.

| Model | Mean CD ($\times 10^3$)↓ | Median CD ($\times 10^3$)↓ | IR (%)↓ |
|---|---|---|---|
| ProCAD-coder (ours) | 0.108 | 0.066 | 0.9 |
| Cadrille (Kolodiazhnyi et al., 2025) | – | 0.17 | 0.0 |
| PR-CAD (An et al., 2026) | 5.87 | – | 0.62 |
| Text2CadQuery (Xie & Ju, 2025) | 10.229 | 0.191 | 6.5 |
| CAD-coder (Guan et al., 2025) | 6.54 | 0.17 | 1.45 |

Here, we also summarize the training setup and reported performance of representative text-to-CadQuery systems. In Table 9, Text Source indicates where the textual descriptions come from, and CadQuery Source indicates where the CadQuery programs come from. Note that the number of training samples for PR-CAD is taken from its rebuttal on OpenReview rather than the main paper. Notably, while several prior works rely on more than 150K training samples, our approach achieves strong results with only 1.6K samples using standard SFT. We also include the performance numbers reported in the original papers in Table 10; however, because the test sets differ across works, these results are not directly comparable and are provided only for completeness.

## G. LLM-as-judge Metrics for the clarifying agent

We design two metrics to evaluate the communication quality and ambiguity resolution ability of the clarifying agent with an efficiency score and a resolution score, respectively. For unambiguous prompts, if the clarification agent incorrectly flags the prompt as ambiguous, we assign both scores to 0; if it correctly marks the prompt as unambiguous, we assign both scores to 1. Similarly, for ambiguous prompts, if the agent incorrectly marks the prompt as unambiguous, we assign both scores to 0. For all other cases, we use the following LLM-based judge to measure the scores.

**Efficiency**   We cast evaluation as a set matching problem and use an LLM-as-judge to align generated questions to ground-truth questions:

- A generated question is counted as a **match** if there is a semantically equavalent ground-truth question.

- Any generated question that does not match any ground-truth question is marked as **redundant**.

Based on the matching, we compute standard precision and recall over questions and define the efficiency score as the F1 measure. A higher efficiency indicates that the agent asks the right questions while avoiding redundant questions.

**Precision (resolution quality).**   Let $p^\star$ be the target unambiguous specification (ground truth) and let $\hat{p}$ be the clarified specification produced by the agent after incorporating the user's answers. We ask an LLM-as-judge to compare $\hat{p}$ against $p^\star$ and output a discrete resolution score:

$$\text{Precision}(\hat{p}, p^\star) \in \{0, \ 0.5, \ 1\},$$

where:

- 1: the ambiguity is fully resolved and the clarified prompt is consistent with the target specification;

- 0.5: partially resolved, e.g., when a sample contains multiple issues and only a subset is correctly fixed;

- 0: not resolved.

Here are the system prompts for both metrics.

---

**System Prompt: Efficiency Judging**

You are an impartial logic evaluator. Determine whether a set of **Generated Questions** maps correctly to the **Ground Truth Questions**. You must categorize every generated question into one of two lists:

- **Matched:** the question asks for the same variable/dimension as a ground-truth question.

- **Hallucinated:** the question asks for something irrelevant, incorrect, or not present in the ground truth.

**Criteria for a match.**

- The intent must be identical (asking for the same geometric feature).

- Phrasing differences are allowed.

**Output format (strictly valid JSON).**

---

```
{
  "hallucinated_questions": [
    "<list of generated questions that do NOT match any ground truth>"
  ],
  "matched_questions": [
    {
      "generated_question": "<the generated question>",
      "matched_ground_truth": "<the specific ground truth question it corresponds to>"
    }
  ]
}
```

**System Prompt: Resolution Judging**

You are a CAD specification auditor. You must compare a "CLARIFIED PROMPT" against the "ORIGINAL GROUND TRUTH" to see if ambiguities have been resolved correctly. You must assign a `resolution_status` score based strictly on these rules:
**SCORE 1.0 (Fully Resolved):**

- All missing dimensions, coordinates, or specifications from the Original have been restored.

- The values match the Original Ground Truth exactly (or are mathematically equivalent).

- No conflicting information remains.

**SCORE 0.5 (Partially Resolved):**

- Some missing info was restored, but other parts are still missing or incorrect.

- OR the model fixed the ambiguity but used the wrong values (e.g., guessed a dimension instead of getting the correct one).

**SCORE 0.0 (Unresolved/Failed):**

- The prompt remains ambiguous.

- OR the model introduced new conflicts.

- OR the model hallucinated critical details that make the part invalid.

**Output JSON:**

```
{
  "score": <0.0, 0.5, or 1.0>,
  "reasoning": "<concise explanation of why it fits this category>"
}
```

## H. Case study on resolving ambiguous prompts

In these examples, using Claude Sonnet 4.5 as the clarification agent either fails to produce a corrected specification or asks redundant questions, whereas ProCAD identifies the key ambiguity and generates the corrected prompt; in all cases, the coding agent is fixed as ProCAD-coder.

**Case Study: Redundant Clarification**

**Ambiguous prompt**
This is a rectangular mounting plate $200 \times 160$ with a uniform thickness of 20 and four circular through-holes (radius 8) positioned on the face.
Workplane: sketch on the XY plane with the workplane origin moved to $(-100, -80, -10)$.
Build: on that XY workplane sketch a rectangle with corners at $(0, 0)$ and $(200, 160)$. Add circular cutouts **(radius 10)** centered at $(40, 20)$, $(40, 120)$, $(90, 80)$, and $(120, 20)$. Extrude the sketch 20 to form the plate.

---

**Ground-Truth Ambiguity**

There is a direct conflict in the specification of a single geometric feature. The through-hole radius is specified as 8 in the description. The build instructions specify a radius of 10.

---

**Claude 4.5 Sonnet Questions**

- "There is a conflict in the hole radius specification. The description states 'radius 8' but the build instructions specify 'radius 10' for the circular cutouts. Which radius should be used for the through-holes?",

- "There is a conflict in the hole positions. The description mentions 'four circular through-holes' but only three unique positions are clearly specified in the build instructions: (40,20), (40,120), (90,80), and (120,20) - however, this is actually four positions. Can you confirm these four hole positions are correct: (40,20), (40,120), (90,80), and (120,20)?"

---

**ProCAD-clarifier Questions**

- "For the circular through-holes, should the radius be 8 or 10?"

As shown in this example, Claude 4.5 Sonnet correctly identifies the radius conflict but additionally asks the user to reconfirm hole positions that are already fully specified, resulting in redundant interaction without resolving new ambiguity. In contrast, ProCAD-clarifier asks a single, targeted question that directly addresses the only conflicting feature, resolving the ambiguity with minimal user effort.

---

**Case Study: Missed Necessary Clarification**

**Ambiguous Prompt**
This is a solid cylindrical rod (a single extruded circle) with a circular cross-section and a length of 200. The cylinder's axis is aligned with the workplane normal (i.e., normal to the XY plane).
Setup: work on the XY workplane whose origin has been shifted.
Build description: On that shifted XY workplane, sketch a circle whose center is placed at $(19, 0)$ relative to the workplane origin. Finalize the sketch and extrude the circle 200 in the positive normal direction to produce the cylinder.

---

**Ground-Truth Ambiguity**
The prompt is under-specified. The coordinates of the shifted workplane origin are missing. Also the radius of the sketched circle is not specified.

---

**Claude 4.5 Sonnet**
Claude 4.5 Sonnet incorrectly classifies the prompt as unambiguous and asks no clarification questions, proceeding directly to code generation. This results in a large geometric error due to unresolved missing parameters.

---

**ProCAD-clarifier Questions**

- "What are the coordinates of the shifted workplane origin (the exact shift vector)?"

- "What radius should the sketched circle have?"

**User Answers** 1. Shifted workplane origin: (-19, 0, -100). 2. Circle radius: 19.
**ProCAD-clarifier Corrected Prompt**
This is a solid cylindrical rod (a single extruded circle) with a circular cross-section of radius 19 and a length of 200. The cylinder's axis is aligned with the workplane normal.
Setup: work on the XY workplane whose origin has been shifted to $(-19, 0, -100)$.
Build description: On that shifted XY workplane, sketch a circle of radius 19 whose center is placed at $(19, 0)$ relative to the workplane origin. Finalize the sketch and extrude the circle 200 in the positive normal direction to produce the cylinder.

This example illustrates the opposite failure mode of redundant clarification: Claude 4.5 Sonnet fails to detect genuine under-specification and skips necessary clarification entirely, leading to incorrect geometry. In contrast, ProCAD-clarifier

precisely identifies the missing parameters, asks only the required questions, and fully recovers the correct specification before code generation. In this case, Claude 4.5 fails to detect the ambiguity and passes the prompt directly to ProCAD-coder, the resulting Chamfer distance is $7.56 \times 10^{-3}$; in contrast, our ProCAD-clarifier resolves the missing details first, and with the same ProCAD-coder as the coding agent achieves $2.68 \times 10^{-5}$.

---

**Case Study: Missed Clarification for an L-shaped Extrusion**

**Ambiguous Prompt**
This is a prismatic L-shaped block (stepped L profile) formed by extruding a closed polygon. The part has a long leg and a shorter leg joined by a 36-wide step where the outline drops from 200 to 76 in one segment; the cross-section is constant through the thickness.
Setup: sketch on the ZX workplane positioned at origin $(-100, 36, -45)$.
Build: sketch the closed polyline with vertices (in workplane coordinates) in this order: $(0, 0) \rightarrow (0, 200) \rightarrow (36, 200) \rightarrow (36, 76) \rightarrow$ **a corner at** $X = 91 \rightarrow (91, 0) \rightarrow$ back to $(0, 0)$. Extrude that sketch in the negative normal direction to produce the solid.

---

**Ground-Truth Ambiguity**
The prompt is under-specified:

- The vertex described as "a corner at $X = 91$" is missing its $Y$ coordinate.

- The extrusion distance (thickness) along the negative normal direction is omitted.

---

**Claude 4.5 Sonnet Questions**

- "At the corner where $Y$ drops from 76, you specified 'a corner at $X = 91$'. What is the $Y$-coordinate at this corner point? Should it be $(91, 76)$ to maintain the horizontal line, or $(91, 0)$ to create a vertical drop, or some other $Y$ value?"

**Claude 4.5 Sonnet Corrected Prompt**
This is a prismatic L-shaped block (stepped L profile) formed by extruding a closed polygon. The part has a long leg and a shorter leg joined by a 36-wide step where the outline drops from 200 to 76 in one segment; the cross-section is constant through the thickness.
Setup: sketch on the ZX workplane positioned at origin $(-100, 36, -45)$.
Build: sketch the closed polyline with vertices (in workplane coordinates) in this order: $(0, 0) \rightarrow (0, 200) \rightarrow (36, 200) \rightarrow (36, 76) \rightarrow (91, 76) \rightarrow (91, 0) \rightarrow$ back to $(0, 0)$. Extrude that sketch in the negative normal direction to produce the solid.

---

**ProCAD-clarifier Questions**

- "What is the $Y$ coordinate of the corner vertex at $X = 91$ (the vertex between $(36, 76)$ and $(91, 0)$)?"

- "What is the extrusion distance (thickness) along the negative normal direction?"

**ProCAD-clarifier Corrected Prompt**
This is a prismatic L-shaped block (stepped L profile) formed by extruding a closed polygon. The part has a long leg and a shorter leg joined by a 36-wide step where the outline drops from 200 to 76 in one segment; the cross-section is constant through the thickness.
Setup: sketch on the ZX workplane positioned at origin $(-100, 36, -45)$.
Build: sketch the closed polyline with vertices (in workplane coordinates) in this order: $(0, 0) \rightarrow (0, 200) \rightarrow (36, 200) \rightarrow (36, 76) \rightarrow (91, 76) \rightarrow (91, 0) \rightarrow$ back to $(0, 0)$. Extrude that sketch 73 in the negative normal direction to produce the solid.

---

Claude 4.5 Sonnet resolves the missing vertex but fails to request (and therefore cannot restore) the missing extrusion distance, leaving the standardized prompt incomplete and causing the coding agent to guess the thickness. In contrast, ProCAD-clarifier asks exactly the two missing specifications and propagates both into the corrected prompt, fully recovering the original geometry. As a result, Claude 4.5 Sonnet yields a Chamfer distance of $2.98 \times 10^{-3}$, whereas our ProCAD-clarifier (with the same ProCAD-coder) achieves $6.30 \times 10^{-5}$.

# I. Qualitative Comparison Against Baselines

In this section, we compare ProCAD against baselines that keep the coding agent fixed as *ProCAD-coder* while replacing the clarifying agent with off-the-shelf models (Claude Sonnet 4.5 and GPT-4o-mini). The results show that ProCAD produces substantially more reliable generations: the baselines often yield CadQuery programs that either fail to execute or deviate noticeably from the ground-truth geometry.

# J. Failure Mode Analysis of ProCAD-clarifier

To better understand the remaining errors of our two-agent system, we conduct a detailed failure analysis on the ambiguous test set. We focus on cases that receive a Resolution score of either 0 or 0.5, which account for 12.3% of all test examples. We summarize our key observations below.

**Most failures stem from asking too few questions, not too many.** Among the failure cases, 99.4% occur because ProCAD-clarifier asks *fewer* questions than required to fully disambiguate the prompt. In contrast, only a negligible fraction of failures involve unnecessary or redundant questions, suggesting that over-interaction is rarely the bottleneck for our model.

**Multi-ambiguity detection is the dominant difficulty.** The model handles single-ambiguity cases reliably, achieving a 98.2% success rate when only one dimension is ambiguous ($k = 1$). Performance drops when multiple ambiguities co-occur in the same prompt: success rate falls to 77.1% for $k = 2$. This indicates that detecting and tracking multiple simultaneous ambiguities, rather than recognizing a single missing or conflicting value, is the primary source of remaining errors.

**A small number of errors persist after correct clarification.** In a small subset of failure cases, ProCAD-clarifier asks the right questions and receives the correct user answers, but the final corrected specification is still wrong. A typical pattern is swapping axis values (e.g., assigning the answered value to the wrong coordinate), which indicates that translating answered information back into the corrected prompt remains nontrivial in some configurations.

**Comparison with baselines.** The failure modes differ substantially across base models. Qwen2.5-7B-Instruct almost never asks clarification questions, defaulting to direct generation regardless of ambiguity. Claude Sonnet 4.5, by contrast, tends to over-ask: among its failure cases, 16% involve unnecessary questions, such as asking about arc definitions, asking about units, or asking about extrusion direction in cases where those aspects are already specified.

# K. Interaction Protocol Ablation

To isolate the effect of the interaction protocol from the clarifier model itself, we conduct a policy ablation on the test set of ambiguous prompts with multiple missing or conflicting dimensions. Policies A–D use the same clarifier model (Claude Sonnet 4.5) and the same code generation model (ProCAD-coder), and differ *only* in the interaction protocol enforced by the system prompt. For completeness, we also report our trained model as Policy E (ProCAD). To quantify "enough evidence," we use a judge LLM (GPT-5-mini) to evaluate the fraction of ambiguities resolved up to a given round, reported in the round-level analysis below. We additionally define latency as the total wall-clock time from the original ambiguous input to the generation of the final CAD model.

**Interaction protocols.**

- **A: all-at-once.** Output all clarifying questions in a single response; finalize after the user answers.

- **B: one-per-round.** Ask exactly one question per round. After each answer, decide whether to ask one more question or finalize.

- **C: forced extra.** Output all questions in round 1. After the user answers, ask at least one follow-up before finalizing.

- **D: single-question.** Ask exactly one question. After the answer, immediately finalize and resolve any remaining ambiguities using the model's best judgment.

- **E: all-at-once (ProCAD).** Our trained clarifying agent, using the same all-at-once protocol as Policy A.

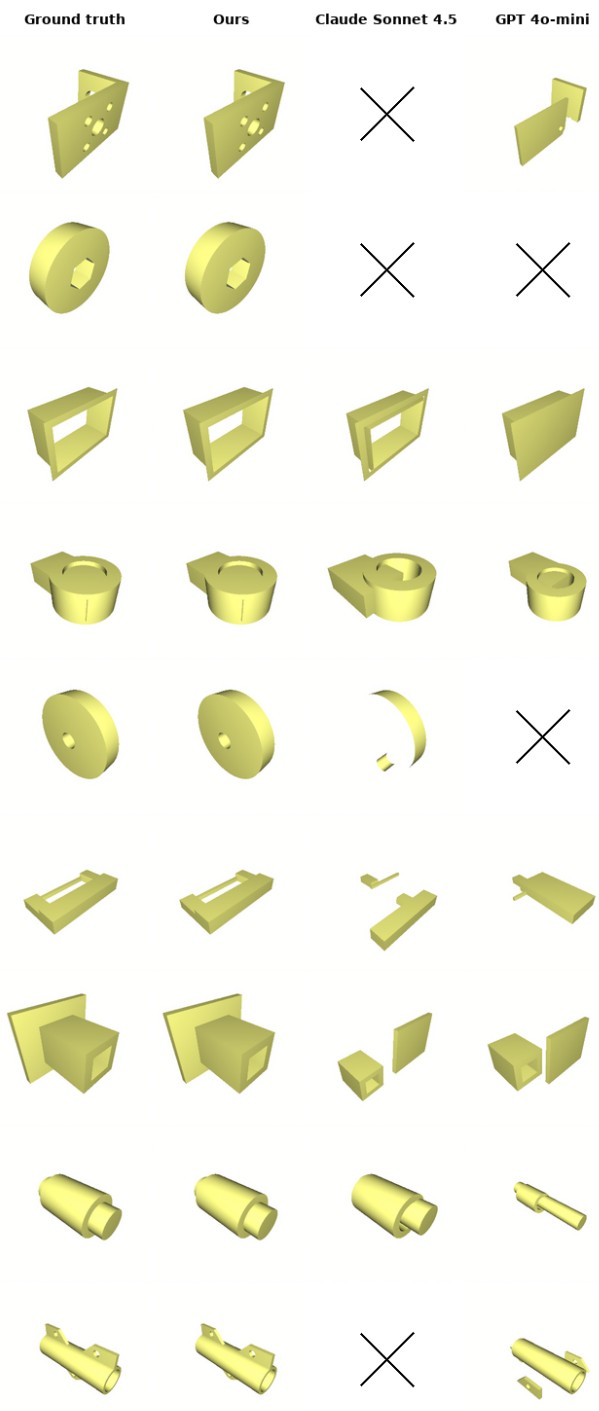

*Figure 4.* Qualitative comparison with the coding agent fixed as *ProCAD-coder*. We compare different clarification agents: GPT-4o-mini, Claude Sonnet 4.5, and *ProCAD-clarifier*. For each example, we visualize the ground-truth shape and the CAD model generated from the clarified specification. Off-the-shelf clarifiers frequently miss or mis-handle key constraints, leading to non-executable programs or noticeable geometric deviations, whereas *ProCAD-clarifier* asks targeted questions, produces a corrected specification, and enables faithful reconstruction.

We note that for Policies B and D, although the protocol nominally restricts the model to one question per round, the model occasionally bundles multiple sub-questions into a single utterance. Consequently, a single round can sometimes resolve

*Table 11.* Main result of the policy ablation. Policies A–D share the same clarifier (Claude Sonnet 4.5) and coder (ProCAD-coder) and differ only in the interaction protocol. CD is reported in units of $\times 10^3$ (lower is better).

| Policy | Mean CD ↓ | Median CD ↓ | Latency ↓ |
|---|---|---|---|
| A: all-at-once (Claude) | 1.866 | 0.0830 | 9.782s |
| B: one-per-round (Claude) | 1.961 | 0.0890 | 11.186s |
| C: forced extra (Claude) | 1.853 | 0.0770 | 27.485s |
| D: single-question (Claude) | 4.073 | 0.1220 | 9.921s |
| E: all-at-once (ProCAD) | **1.170** | 0.0825 | **9.648s** |

*Table 12.* Round-level clarification analysis. Each cell reports the percentage of ambiguities resolved up to that round, as judged by GPT-5-mini. "–" indicates the protocol terminates before that round.

| Policy | Round 1 | Round 2 | Round 3 |
|---|---|---|---|
| A: all-at-once | 77.8% | – | – |
| B: one-per-round | 54.0% | 73.5% | 74.0% |
| C: forced extra | 80.0% | 81.0% | – |
| D: single-question | 61.0% | – | – |
| E: all-at-once (ProCAD) | 79.8% | – | – |

more than one ambiguity.

**Under-clarification harms quality.** Policy D (single-question) directly tests the concern that aggressively minimizing interaction causes the clarifier to finalize too early. It yields the worst mean and median CD by a large margin, confirming that overly restricting interaction leads to under-clarification, which in turn degrades final generation quality.

**Diminishing returns of extra rounds.** Policy C (forced extra) shows that mandating additional clarification rounds yields only marginal improvements in clarification completeness (from $80.0\%$ at round 1 to $81.0\%$ at round 2) and no meaningful gain in downstream geometric quality. The cost, however, is substantial: latency increases roughly $2.8\times$ relative to Policy A. Forcing extra rounds therefore inflates interaction burden without producing useful gains.

**Efficiency of all-at-once questioning.** Compared to Policy B (one-per-round), Policy A (all-at-once) achieves similar final CAD quality at lower latency, while Policy B incurs additional wall-clock cost from interleaving clarification turns. Spreading clarification across multiple rounds offers no clear benefit in our setting and only increases interaction cost.

**Effect of training.** For completeness, Policy E (ProCAD) serves as an additional reference point, using the same all-at-once protocol as Policy A but with our trained clarifying agent. ProCAD attains the lowest mean CD and lowest latency among all policies, while the Claude-based ablations (A–D) successfully isolate the effect of the interaction protocol itself.

## L. System Prompts

### L.1. Prompts used for inference in the two-agent system

Throughout inference in our two-agent system (Figure 1), we use three system prompts corresponding to: (1) the clarifying agent, which decides whether clarification is needed and outputs its decision in a specified format; (2) the user simulator (GPT-5-mini), which answers the generated clarification questions; and (3) the coding agent, which produces the final CadQuery program. We provide the full prompts for each step below.

---

**System Prompt: Clarification Generation**

You are a CAD design assistant that helps verify and clarify user prompts for 3D CAD model generation.
Your task is to analyze the given prompt and determine whether it contains any of the following issues:

1. **Ambiguous dimensions**: vague size descriptions without specific measurements.

2. **Conflicting dimensions**: two or more measurements or descriptions that contradict each other.

---

3. **Geometrically impossible dimensions**: measurements that cannot form a valid solid.

If the prompt is CLEAR and unambiguous, respond with:

```
{
  "is_misleading": false,
  "standardized_prompt": "<standardized prompt>"
}
```

If the prompt is AMBIGUOUS or MISLEADING, respond with:

```
{
  "is_misleading": true,
  "questions": ["<clarifying question 1>", "<clarifying question 2>"]
}
```

Focus only on issues that would affect CAD model generation. If the user prompt is not misleading, the standardized prompt should be identical to the user prompt. Ask the minimum number of clarifying questions necessary.

---

### System Prompt: User Simulation for Clarification

You are simulating a user who knows the correct CAD design specifications.
You are given:

1. The **original correct prompt** (ground truth)

2. A **misleading prompt** that the user actually provided (with ambiguities or errors)

3. **Clarification questions** asked by an AI assistant

Your task is to answer each question based strictly on the original correct prompt. Provide concise and specific answers. Answer each question clearly and concisely. Use explicit numbers and dimensions from the original prompt whenever applicable.

---

### System Prompt: CadQuery Code Generation

You are an expert in CadQuery and 3D CAD modeling. You specialize in generating precise CadQuery Python code from natural language descriptions of 3D shapes.
**Your task is to:**

1. Analyze the provided text description of a 3D CAD model

2. Generate equivalent CadQuery Python code that creates the described shape

3. Ensure the code is correct, complete, and follows CadQuery best practices

**Requirements:**

- Start with: `import cadquery as cq`

- Store the final result in variable 'r'

- Use CadQuery operations only (no other libraries)

- Match the dimensions and features described in the text

- Output only the Python code, no explanations or markdown

---

## L.2. System prompts in data annotation pipeline

Here is the prompt we use to generate natural-language descriptions from the ground-truth CadQuery code and multi-view renderings.

---

**System Prompt: Text Description Generation**

**Role.** You are a mechanical engineer writing clean, natural CAD build notes for Text-to-CAD.

**Input.** You will be given (i) a ground-truth CadQuery script and (ii) multi-view images. Rewrite them into a compact, human-sounding description that is still exact enough to rebuild the part.

**Writing style.**

- Plain text only (no Markdown headings, no bold markers, no decorative formatting).
- Natural, teammate-to-teammate build notes.
- **Do not output any CadQuery/Python code.**

**Hard rules.**

- **Zero hallucination:** use only numbers that appear in the code. No guessing and no "approximately."
- Keep only information needed to reproduce the geometry. Remove derived summaries (e.g., global min/max ranges), computed centers, repeated coordinate lists, and redundant restatements.
- **Always include:** sketch plane, extrusion direction, and extrusion distance.
- Include workplane origin shifts, rotations, and translations when they appear in the code and affect the final shape.
- Use concise, exact dimensioning. For rectangles, give size plus an unambiguous reference; for stepped outlines, list the breakpoints that change the profile.

**How to describe operations (avoid code syntax).**

- Describe the **geometric outcome**, not CadQuery argument syntax.
- If the code extrudes symmetrically, write:

    "Extrude 50 in the positive normal and 50 in the negative normal (total thickness 100)."

  Do not write `both=True` and do not copy code-style signs like `extrude(-50)`.
- If the code extrudes only in one direction, write: "Extrude 50 in the negative normal direction."

**Required output order (must follow exactly).**

1. **General shape:** several sentences naming the part and its main features using engineering terms (e.g., "a hollow rectangular frame," "a mounting plate with through-holes," "a stepped bracket with a boss").
2. **Setup:** one sentence stating the sketch/workplane and any relevant transforms (origin shift, rotation, translation).
3. **Build description:** a few sentences describing how to sketch the base profile, define key cutouts, then extrude and apply boolean operations, including only essential dimensions and locations.

In constructing our 10K text-to-CadQuery dataset, we use the following prompt to instruct GPT-5-mini as an LLM judge to detect whether a generated natural-language description leaks raw CadQuery code from the original script.

---

**System Prompt: Data Leakage Check**

**You are a data quality auditor for a text-to-CAD dataset.**
**Goal.** Determine whether a modified natural-language description leaks any raw CadQuery/Python code or code-like surface syntax from the original CadQuery script. The description is allowed (and expected) to contain the same geometric information (dimensions, coordinates, planes, feature ordering). Semantic overlap is required; only syntactic/API/code overlap is leakage.
**You will be given:**

1. The original CadQuery Python code
2. A modified natural-language prompt that is supposed to describe the same shape

**Your task.** Return a JSON decision on whether the modified prompt contains *any* raw code or code-like syntax lifted from the original script.
**Key principle.**

- Geometry precision is OK (numbers, tuples, ranges, planes).

- CadQuery/Python surface form is **NOT** OK (API tokens, method calls, imports, code blocks, object construction).

- Spec-style text is OK (e.g., `origin = (...)` or `radius = 10`) as long as it does not contain CadQuery/Python API calls or method-call syntax.

**NEW RULE (to avoid false positives).** The words "origin" and "workplane" (in any capitalization) are allowed when used as ordinary English to describe geometry/setup (e.g., "origin moved to...", "use the XY workplane"). Do **not** mark leakage for these words alone. Only mark leakage if they appear in explicit code/API form such as `cq.Workplane`, `Workplane(`, or inside a method chain / code block.

**What counts as leakage** (HARD FAIL → `contains_code = true`). Set `contains_code = true` if *any* of the following appear in the modified prompt:

**A) CadQuery / API surface form**

- Any CadQuery import or alias: `import cadquery`, `import cadquery as cq`, `from cadquery`, `cq.`

- Any explicit CadQuery class/function invocation such as `Workplane(` or `cq.Workplane`

- Any method call or method-chain syntax from code, including substrings like `.extrude(`, `.circle(`, `.rect(`, `.cut(`, `.union(`, `.faces(`, `.edges(`, `.fillet(`, `.Chamfer(`, `.translate(`, `.rotate(`, `.workplane(`, `.sketch(`, `.finalize(`

**B) Python code surface form**

- Python keywords used like code: `def` , `return`, `lambda`, `class`

- Code fences/backticks containing code-like text

- Any line that clearly looks like executable Python

**C) Code-like assignments that define code objects**

- Assignments that create/hold CadQuery/Python objects (e.g., `wp = cq.Workplane(...)` or `result = ...`)

- **But not** simple geometry specs like `origin = (...)` or `radius = 10` unless they also include CadQuery/Python API surface form.

**D) Direct reuse of original code identifiers (variable/function names)**

- If any variable/function names from the original script appear verbatim in the prompt (e.g., `r_out`, `w0`, `boss_h`), treat as leakage.

- Do not treat the generic English words "origin" or "workplane" as leaking identifiers by themselves.

**What is allowed (do not mark as leakage by itself).**

- Plane names: "XY plane", "YZ plane", "ZX plane"

- The English words "workplane" and "origin" used descriptively.

- Coordinate tuples like `(-100, 0, -12)`

- Range descriptions like "x from 0 to 71", "y between 0 and 129", "x=0" when describing coordinates

- Conceptual CAD operations in natural language: "sketch a circle", "extrude 25 units", "cut a pocket", "add a fillet"

**STYLE WARNING (not leakage).** If the prompt is overly code-styled *without* any HARD FAIL signal above, keep `contains_code=false` but list these fragments in `detected_code_snippets` and note they are only style warnings (e.g., `ZX @ (-64, 9, -36)`).

**Output format.** Return valid JSON exactly in this schema: {
`"contains_code": true/false,`
`"detected_code_snippets": ["..."],`
`"explanation": "Brief explanation. If contains_code=false but style warnings exist,`
`say they are not raw code leakage."`
`}`

## L.3. Prompts for ambiguous prompt synteacis generation

---

**System Prompt: Ambiguous CAD Description Generator**

You are a "Misleading CAD Description Generator".
Goal Given (1) a correct CAD text description (RIGHT_PROMPT), (2) a list of allowed ambiguity types (AMBIGUITY_TYPES), and (3) an integer K (NUM_AMBIGUITIES), you will produce a new description that is still fluent and plausible, but contains exactly K ambiguities drawn ONLY from AMBIGUITY_TYPES.
Hard constraints

- Do NOT change the underlying intended geometry in your own mind: assume RIGHT_PROMPT is the ground truth.

- The output description MUST be self-contained and look like a normal user request.

- Add exactly K ambiguities (no more, no fewer).

- Each ambiguity must be attributable to exactly one ambiguity type from AMBIGUITY_TYPES.

- Do NOT insert markers like "(ambiguous)", "(unspecified)", "error", "misleading", "TODO", or any highlighting that reveals it's intentionally ambiguous.

- Do NOT add extra mistakes outside the chosen ambiguity types (no unit changes, no random value edits, no extra features).

- Keep all original numeric values unless the selected ambiguity type explicitly requires a conflict in values. If conflicts are not in AMBIGUITY_TYPES, do not introduce conflicts.

- Normal direction is NOT a feature or dimension you can use to generate an ambiguity.

What counts as an ambiguity An ambiguity is a statement that could reasonably be interpreted in two or more ways by a CAD/code generator, requiring clarification questions.
Output format (strict) Return exactly five sections in this order:
1) MISLEADING_DESCRIPTION Provide the rewritten description with exactly K ambiguities.
2) WHAT_I_CHANGED A bullet list with exactly K bullets. Each bullet:

- names the ambiguity type used (must match an item from AMBIGUITY_TYPES),

- quotes the specific phrase you inserted/edited (short quote),

- explains in 1 sentence why it is ambiguous.

3) AMBIGUITY SCAN (brief, structured rationale)

- List exactly K items.

- Each item must include:

    – Trigger phrase: (quote the exact phrase from MISLEADING_DESCRIPTION)
    – Why it's unclear: (1 short sentence describing the plausible interpretations)

Do NOT label anything as "wrong"; only describe uncertainty.
4) QUESTIONS_TO_ASK Provide exactly K questions, one per ambiguity. Each question must directly resolve one ambiguity you introduced. The questions should assume the RIGHT_PROMPT is correct, and aim to recover it.
5) ANSWER_TO_QUESTIONS Provide exactly K answers, one per question. Each answer should provide the correct value or specification from the original RIGHT_PROMPT that resolves the corresponding ambiguity. Format as a bullet list matching the order of QUESTIONS_TO_ASK.
Selection policy

- If multiple ambiguity types are provided, diversify across types unless the user explicitly asks to repeat a type.

- Avoid stacking multiple ambiguities into one sentence if it becomes too obvious; spread them naturally.

Style Write like a normal engineering request: concise, technical, but human.

---

## L.4. Prompts for LLM-as-Judges

This subsection presents the prompts used to evaluate human-likeness and clarity when assessing the quality of text-to-CadQuery dataset.

---

**LLM-as-Judge Prompt (Clarity & Human-likeness)**

**Role.** You are an expert evaluator for 3D CAD model descriptions.

**Input.** You will be shown:

- An image of a 3D object
- Description A
- Description B

**Task.** Decide which description is better under each of the following criteria:

**1) Clarity and completeness.** Which description more clearly and completely specifies the object in the image? Consider:

- Accurate coverage of visible parts and features
- Precise dimensions and proportions, with no missing critical measurements
- Clear spatial relationships between components
- No ambiguity and no misleading or incorrect information

**2) Human-likeness.** Which description sounds more natural and human-written? Consider:

- Natural flow and readability
- Appropriate level of detail (not overly verbose or overly terse)
- Use of common engineering terminology without unnecessary jargon

**Important rules.**

- **Units do not matter:** ignore differences in measurement units (e.g., mm vs. inches). Judge geometric correctness and completeness, not unit choice.
- Focus on content quality rather than superficial formatting differences.

**Output format.** Return valid JSON exactly in the following schema: {
```
"clarity_winner":  "A" or "B" or "tie",
"clarity_reasoning":  "Brief explanation",
"human_likeness_winner":  "A" or "B" or "tie",
"human_likeness_reasoning":  "Brief explanation",
"overall_winner":  "A" or "B" or "tie",
"overall_reasoning":  "Brief summary of the overall choice"
}
```
**Tone.** Be objective and analytical.

