# OpenReview forum: "Clarify Before You Draw: Proactive Agents for Robust Text-to-CAD Generation"
_ICML.cc/2026/Conference — ICML 2026 regular_

### Official Review · Reviewer_RXef · 2026-02-28

**Soundness:** 3
**Presentation:** 2
**Significance:** 1
**Originality:** 2
**Overall Recommendation:** 4
**Confidence:** 3

**Summary:**

This paper is about CAD code (cadquery) generation with 2 agents. Both use model Qwen2.5-7B-Instruct. For the clarifying agent, the input is the possibly ambiguous prompt, and the expected output is the clarification (JSON). For the second code agent, the input is unambiguous natural language, and the output is the code. The training is SFT with 2 datasets.

**Compliance With Llm Reviewing Policy:**

Affirmed.

**Final Justification:**

Please consider incorporating the rebuttal into the final version. I've raised my score.

**Key Questions For Authors:**

- Line 76: Does text2cad (khan2024b) generate a JSON representation instead of CadQuery code?
- Line 72 (right column): “Consequently, current methods XXX …” needs citation. The cited work in 2013 mainly discusses measuring embodiment design characteristics and is not really about "current" methods.
- Line 130: “We also provide one example where ... radius of the inner icicle.”  “icicle” seems to be a typo.
- Line 141: Please explain why introducing a Clarifying Agent mitigates reward hacking, especially when the clarification questions themselves are generated by the Clarifying Agent model, **not produced by the environment/users**.
- Line 140 (right column): Why is C(h) treated as the reward of the clarifying agent, **in order to encourage low interaction burden**? **Isn’t this term itself highly vulnerable to reward hacking?**
- Line 186 (right column): “Compared with the Text2CAD pipeline, our approach offers three key advantages.” needs citation.
- Line 255–260: “We assume that the user can provide correct answers to any asked question as long as the question itself is clear. Under this assumption, an optimal policy should minimize the number of interaction rounds…” I think this statement lacks causal justification. The authors seem to justify minimizing interaction rounds. But I'm afraid that this encourages the model to end the dialogue as quickly as possible. This has nothing to do with whether the environment/users can provide perfect answers.
- The clarifying agent is trained with **single-turn SFT**. though the task involves multi-turn interaction. If possible, a comparison with true multi-turn training should be added.
- Some parts of the paper are not well organized. There is overlap between Section 3 and Section 5.2.

Sorry that my de facto rating is between 2.5 and 3 (but I gave a 3 in overall recommendation). I hope the authors can properly address the points I **highlighted**.

**Limitations:**

yes

**Strengths And Weaknesses:**

Strengths:
- The paper is easy to follow.
- The paper is strong from an engineering perspective.
- They build **new data** and emphasize the data quality.

Weaknesses:
- My biggest concern is that the value to the general AI research community is limited, and it may not meet the bar of icml, though the engineering implementation is good. Decomposing a research task into multiple stages and solving it with multi-agent LLM systems feels more like an engineering paradigm.
- The ambiguity level of the clarifying agent’s input varies. For highly ambiguous inputs, the output should inherently be highly uncertain (e.g., the user’s initial request is very vague). However, the sequential clarification process forces the interaction into a fixed question-answer path, **which ultimately constrains the resulting specification to follow a predetermined trajectory**. This may artificially reduce uncertainty instead of modeling it properly.

---

> ### Author Rebuttal · Authors · 2026-03-31
>
> We sincerely thank the Reviewer for the detailed feedback and recognize our paper as easy-to-follow, strong in engineering, with a high-quality dataset.
>
> **W1**: We respectfully disagree with the characterization of our paper as primarily an engineering implementation. In the LLM-based text-to-CAD literature, most prior work in the related work section focuses on direct one-shot generation with standard SFT or RL. In contrast, our paper is one of **the first** to explicitly formulate text-to-CAD under ambiguous specifications and to introduce an agentic framework together with a finite-horizon Markov decision process formulation. The contribution is therefore not only a system design, but also a novel problem formulation for reliable CAD generation under ambiguity.
>
> In addition, the paper contributes a broader conceptual message for this research direction. As shown in Tables 1, 8 and 9, our results suggest that progress in text-to-CAD is not only about scaling the computation, but also about improving the quality of the text specification itself and explicitly resolving ambiguity before code generation.
>
> Moreover, we would like to emphasize a key distinction between text-to-CAD and other generation tasks such as text-to-image or text-to-video. For the latter, ambiguity in the prompt is often acceptable, and diversity in the outputs can even be desirable. In CAD design, however, the generated model is typically intended for downstream engineering, where even small dimensional inaccuracies may render the design invalid. This makes the problem fundamentally different from the well-studied LLM applications.
>
> **W2**: In CAD, especially CAD software, unlike image or video generation, the end product is not intended to represent one plausible interpretation among many. For CAD practitioners, it should be a fully specified design that can support downstream engineering and manufacturing. Prior work on parametric CAD also emphasizes the importance of fully defined sketches (e.g., Casey et al., ICCV 2025). Without explicit clarification, previous text-to-CAD models still tend to produce a fully defined CAD model, which means that any ambiguous parts are often filled in by the **LLM’s hallucinations**. So even if the user intent is highly uncertain, a reliable CAD system should explicitly identify what is ambiguous and what must be finally defined, which is the core ability of our clarifying agent.
>
> **Q1**: No. Text2CAD does not generate JSON or CadQuery code. It is a text-to-CAD dataset whose text descriptions are from the minimal JSON specifications in DeepCAD. Text2CAD uses descriptions from minimal JSON with LLMs but without human verification, while our dataset uses a semi-automatic pipeline with human and LLM verification, as shown in Figure 2.
>
> **Q4**: Our intention was not to claim that clarification formally eliminates reward hacking. Rather, the clarifying agent introduces an explicit intermediate specification grounded in user feedback, making the generation process more interpretable and controllable. We will revise this sentence accordingly without the need to change the rest of the paper.
>
> **Q5**: Our intention was not to claim that C(h) by itself is a robust standalone reward. Rather, C(h) is introduced as part of a conceptual trade-off: the clarifying agent should improve downstream geometric fidelity while keeping user interaction minimal. If the agent tries to obtain a higher reward simply by asking fewer questions, it will typically fail to gather enough information to recover the correct CAD model, and this will be reflected in a worse Chamfer Distance.
>
> **Q7**: Thank you for pointing this out. Our intended claim is not that the agent should terminate the dialogue as quickly as possible. Rather, under our assumption that users can provide correct answers once the questions are clear, if the agent has already identified all missing or conflicting constraints, the optimal strategy is to ask all necessary questions in one round. In that case, additional rounds do not provide extra information and only increase interaction burden.
>
> **Q8**: In our current setting, the clarifying agent is not intended to have arbitrary long-horizon dialogue. Instead, under our assumption that users can provide correct answers once the questions are clear, we deliberately reduce the interaction to a two-round policy: the agent first either accepts the prompt or asks a set of targeted questions in one message, and after receiving the answers, it outputs the corrected specification. We agree that true multi-turn training would be interesting. This requires collecting large-scale user data, such as partial, incorrect, or inconsistent answers. It is beyond the scope of the current paper, and we plan to address this in the following work.
>
> We thank reviewer for other questions regarding typos, missing citations and overlap between sections, we will address them accordingly in the revision.

---

> > ### Author Rebuttal · Reviewer_RXef · 2026-04-02
> >
> > > If the agent tries to obtain a higher reward simply by asking fewer questions, it will typically fail to gather enough information to recover the correct CAD model, and this will be reflected in a worse Chamfer Distance.
> >
> > My concern is still lingering on Q4 and Q5. Sorry I'm still not convinced. Could the authors provide experimental evidence to justify this claim?
> >
> > > In that case, additional rounds do not provide extra information and only increase interaction burden.
> >
> > Could you give a more quantifiable metric to define what **enough evidence** is and compare your policy with *non-early-stop* policy?
> >
> > I appreciate your efforts in updating the results.

---

> > > ### Author Response · Authors · 2026-04-02
> > >
> > > **New Response**: Since today is the last day of the discussion, we sincerely hope to check back with the reviewer if our response to your questions is satisfactory or if we can provide further explanations.
> > >
> > > ---
> > >
> > > Thank you for the thoughtful follow-up. We agree that our original wording regarding “reward hacking” was confusing. Our goal is to study a practical trade-off: the clarifier must gather sufficient information to recover a complete CAD specification while avoiding unnecessary interaction overhead. Accordingly, we will revise Section 3 to remove the “reward hacking” framing and clarify that $C(h)$ is strictly a conceptual interaction-cost term, not a standalone robustness claim.
> > >
> > > To further isolate the effect of the interaction protocol, we have added new policy ablations evaluated on the test set on ambiguous prompts with multiple missing or conflicting dimensions. Policies A, B, C, and D all use the same clarifier model (Claude Sonnet 4.5) and the same code generation model (ProCAD-coder). They differ solely in the interaction protocol enforced by the system prompt. For completeness, we also report our trained model (Policy E: ProCAD). To quantify “**enough evidence**,” we use a judge LLM (gpt-5-mini) to **evaluate the percentage of ambiguities resolved up to a given round**, which we report in our Round-Level Clarification Analysis. Additionally, we define **latency** as the total time from the original ambiguous input to the generation of the final CAD model.
> > >
> > > The interaction protocols are defined as follows:
> > >
> > > - **A: all-at-once:** Output all clarifying questions in a single response. Finalize after the user answers.
> > > - **B: one-per-round:** Ask exactly one question per round. After each answer, decide whether to ask one more question or finalize.
> > > - **C: forced extra:** Output all questions in round 1. After the user answers, ask at least one follow-up before finalizing.
> > > - **D: single-question:** Ask exactly one question. After the answer, immediately finalize and resolve any remaining ambiguities using best judgment.
> > >
> > > ### Main Result Table
> > >
> > > | Policy | Mean CD ($10^3$) ↓ | Median CD ($10^3$) ↓ | Latency ↓ |
> > > | --- | --- | --- | --- |
> > > | **A: all-at-once** (Claude) | 1.866 | 0.0830 | 9.782s |
> > > | **B: one-per-round** (Claude) | 1.961 | 0.0890 | 11.186s |
> > > | **C: forced extra** (Claude) | 1.853 | 0.0770 | 27.485s |
> > > | **D: single-question** (Claude) | 4.073 | 0.122 | 9.921s |
> > > | **E: all-at-once** (ProCAD) | 1.170 | 0.0825 | 9.648s |
> > >
> > > ### Round-Level Clarification Analysis
> > >
> > > | Policy | Round 1 | Round 2 | Round 3 |
> > > | --- | --- | --- | --- |
> > > | **A: all-at-once** | 77.8% | - | - |
> > > | **B: one-per-round** | 54.0% | 73.5% | 74.0% |
> > > | **C: forced extra** | 80.0% | 81.0% | - |
> > > | **D: single-question** | 61.0% | - | - |
> > > | **E: all-at-once** | 79.8% | - | - |
> > >
> > > We note that for Policies B and D, although the protocol nominally restricts the model to one question per round, the model occasionally bundles multiple sub-questions into a single one. Consequently, a single round can sometimes resolve more than one ambiguity.
> > >
> > > Our findings highlight the following:
> > >
> > > 1. **Under-clarification harms quality:** The single-question baseline (Policy D) directly tests the concern that aggressively minimizing interaction causes the clarifier to finalize too early. The results confirm that overly restricting interaction leads to under-clarification, which directly degrades final generation quality.
> > > 2. **Diminishing returns of extra rounds:** The forced-extra baseline (Policy C) demonstrates that additional rounds yield only marginal improvements in clarification completeness. This approach significantly increases interaction burden and latency without generating meaningful gains in downstream CAD quality.
> > > 3. **Efficiency of all-at-once questioning:** Compared to the one-per-round approach (Policy B), our all-at-once policy (Policy A) is highly interaction-efficient. While the final CAD quality is broadly similar, the interleaved policy incurs higher latency. Spreading clarification across multiple rounds offers no clear benefit here and only increases interaction costs.
> > >
> > > For completeness, we also include our trained model (ProCAD) as an additional reference point. It remains a strong overall baseline, while the new Claude-based ablations successfully isolate the impact of the interaction protocol itself.
> > >
> > > Taken together, we do not claim that our formulation formally prevents reward hacking. Rather, the empirical evidence demonstrates that:
> > >
> > > - Asking too few questions harms quality;
> > > - Forcing additional rounds yields only modest gains at a high latency cost; and
> > > - Our all-at-once formulation is a highly practical, efficiency-oriented design choice that strikes a strong balance between clarification quality and interaction burden.
> > >
> > > Thank you again for your constructive feedback. We hope these revised clarifications and new ablations make the scope of our claims much more precise.

---

### Official Review · Reviewer_SEPx · 2026-03-11

**Soundness:** 3
**Presentation:** 3
**Significance:** 3
**Originality:** 3
**Overall Recommendation:** 4
**Confidence:** 5

**Summary:**

This addresses a fundamental gap in text-to-CAD generation where existing systems blindly follow ambiguous or underspecified user prompts and hallucinate missing dimensions. The authors propose a two-agent framework pairing a proactive clarifying agent that detects missing or conflicting geometric constraints and asks targeted questions, with a coding agent that synthesizes executable CadQuery programs from the resolved specification. A new high-quality 10K text-to-CadQuery dataset is constructed using a semi-automatic pipeline with leakage and completeness verification.

**Compliance With Llm Reviewing Policy:**

Affirmed.

**Final Justification:**

My primary concerns have been addressed.

**Key Questions For Authors:**

- In the text-to-CAD domain, Chamfer Distance isn't a reliable way for reward calculation or evaluation. When a user says "generate a cylinder", there is no single correct radius or height. The clarifying agent may recover a radius of 10cm while the ground truth annotation uses 2cm. Both are valid cylinders, yet the Chamfer Distance would be enormous despite the shape being semantically and topologically correct.

- The entire evaluation relies on GPT-5-mini as both the data generator, user simulator, and LLM judge, creating a circular evaluation loop. How do the authors ensure that reported performance gains are not artifacts of this circularity, rather than genuine improvements in CAD generation quality? A response demonstrating evaluation with independent human expert judges would significantly strengthen confidence in the results.

- Did the authors try on free-form prompts where real users interact with the system naturally, without synthetically perturbed specifications? Some easy examples, like the cylinder one with different radius values. How good is the generation agent in that case?

I am at a borderline for this paper. My primary concern is with the practicality of the idea in its current form and experimental analysis. With proper explanation, I can reconsider the score

**Limitations:**

Yes

**Strengths And Weaknesses:**

## Strengths

- The problem is genuinely underexplored. Text-to-CAD is inherently hard because of ambiguity.

- The dataset is novel and could be useful for the community.

- The paper is well written and clear to understand.


## Weakness

Regarding its practicality, ProCAD assumes users can always answer clarification questions correctly, but this creates a logical paradox. If the human user already knows the exact dimensions, they would have simply included them in the prompt. And if they do not know, asking _"What is the radius of the holes?"_ or _"What is the extrusion depth? Dimension of the shape?"_ leaves them completely stuck, defeating the purpose of a natural language interface entirely. The system is therefore only practical for an unrealistically narrow user type, someone who knows all dimensions precisely but simply forgot to mention them. Can the user specify its practical use case?

---

> ### Author Rebuttal · Authors · 2026-03-31
>
> We sincerely thank the Reviewer for the detailed feedback and recognize our paper as well-motivated with a clear presentation and a novel dataset.
>
> **W1**: Our target setting is broader than a user who “already knows all dimensions but simply forgot to mention them.” In practice, we consider at least two realistic cases.
>
> First, the user may know only part of the dimensions, but making the CAD model fully defined still requires additional CAD expertise. The clarification step can still help by explicitly surfacing which constraints must be specified before a fully defined CAD model can be produced. The user may then provide an estimate or a preferred design choice. In this case, even If the user does not have an exact dimension in mind initially, the system does not necessarily stuck.
>
> Second, the user may in fact know the full intended design, but for a complex CAD model it is cumbersome to express every constraint correctly in a single natural-language prompt. The clarifying agent serves as a checking and structuring mechanism: it helps surface omitted or conflicting constraints before code generation. For example, even for a simple shape such as a rectangular mounting plate with four corner holes (Lines 920–925), producing a fully defined CAD model still requires specifying the origin, workplane, plate length and width, hole locations, hole radius, and thickness. For more complex shapes, the number of such constraints becomes much larger. For more examples, please refer to Appendix H.
>
> More importantly, if the user does **not** know the missing dimensions, previous text-to-CAD systems will still silently generate a fully defined CAD model. In such cases, the unspecified parts are filled in by hallucination, which is particularly dangerous in CAD
>
> **Q1**：In our benchmark, each sample is tied to a fixed underlying CAD model. The clarifying agent is evaluated on whether it helps recover the original intended specification. Under this controlled setting, Chamfer Distance is a reasonable metric for geometric fidelity to the target shape. It is also widely used in recent text-to-CAD work, such as cadrille, Seek-CAD, CAD-Tokenizer and CAD-Coder. We will clarify this distinction in the revision and also include additional metrics, such as Coverage, Minimum Matching Distance, and VLM Score.
>
> **Q2**: To address this concern, we added human-centered experiments in which both the ambiguous prompts and the user responses are provided by real annotators rather than LLM simulators. Specifically, we sampled 100 examples from our Text-to-CAD dataset. We provided the example including the original unambiguous prompt with the reference image to human annotators, since the original prompt contains the precise dimensions. We then asked multiple annotators to write ambiguous prompts. The resulting clarification questions and corrected prompts were further evaluated by **CAD experts**.
>
> In this experiment, we fix the downstream generation model as ProCAD-coder for all methods. As shown in the tables below, our method continues to achieve the strongest performance.
>
> | Model | Efficiency | Resolution | Mean CD  | Median CD  | Invalid Ratio  |
> | --- | --- | --- | --- | --- | --- |
> | **ProCAD-clarifier** | 0.760 | 0.787 | 0.00128 | 0.00009 | 12.2% |
> | Claude-sonnet-4-5 | 0.598 | 0.700 | 0.00972 | 0.00011 | 12.2% |
> | GPT-4o-mini | 0.251 | 0.287 | 0.01381 | 0.00014 | 19.5% |
> | Qwen2.5-7B | 0.016 | 0.000 | 0.01398 | 0.00369 | 14.6% |
>
> Moreover, we randomly sampled 100 test examples from the synthetic dataset and compared LLM-based evaluation against human evaluation of interaction quality. The two agree on more than 96% of the samples, which provides additional evidence that the LLM-based metrics are reasonably reliable in our setting.
>
> **Q3**:  As a preliminary check, we manually constructed five simple free-form prompts.
>
> | # | Prompt | Missing |
> | --- | --- | --- |
> | 1 | Create a cylinder with radius 5. | height |
> | 2 | Make a rectangular plate that is 80 long and 50 wide. | thickness |
> | 3 | Create a ring with outer radius 20. | inner radius, thickness |
> | 4 | Make a hollow tube with outer radius 10 and height 50. | inner radius |
> | 5 | Create a cylinder 40 high with radius 8 and a through hole of radius 3. | coordinates of the through-hole center |
>
> On these five examples, our clarifying agent successfully detected all missing information, and the coding agent generated the correct CadQuery program after receiving the clarified prompt. For the clarifying agent, we used in-context examples to specify the expected output format.
>
> For comparison, Claude Sonnet 4.5 correctly identified the ambiguity in the first four examples but missed the last one, because it implicitly assumed that the through-hole was centered in the cylinder. GPT-4o-mini identified 3 out of 5 ambiguities, while Qwen2.5-7B-Instruct, the base model of our clarifying agent, did not detect any of them. We will include more examples in the revision.

---

> > ### Author Rebuttal · Reviewer_SEPx · 2026-04-05
> >
> > My primary concerns have been answered.
> > I have raised the score. Please include the additional points in the final revision.

---

> > > ### Author Response · Authors · 2026-04-05
> > >
> > > We sincerely thank you for the positive feedback and for raising the score. We are glad our response addressed your concerns. We will incorporate the additional details, including human-centered experiments, free-form prompts, and additional metrics, into the revised paper.

---

### Official Review · Reviewer_cqV4 · 2026-03-12

**Soundness:** 3
**Presentation:** 3
**Significance:** 3
**Originality:** 3
**Overall Recommendation:** 4
**Confidence:** 4

**Summary:**

The paper studies the problem of generating parametric CAD programs from natural language descriptions and focuses on the challenge that real-world prompts are often ambiguous, incomplete, or internally inconsistent. The authors propose ProCAD, a proactive agentic framework that improves text-to-CAD generation by resolving specification issues before code synthesis. The system contains two agents: a clarifying agent that analyzes the user prompt and asks targeted clarification questions when important dimensions or constraints are missing, and a coding agent that converts the finalized specification into an executable CadQuery program. The authors proceed to discuss an important domain where natural language descriptions frequently lack precise geometric constraints, which often causes existing text-to-CAD models to hallucinate dimensions or generate invalid code.
The manuscript introduces a curated Text-to-CadQuery dataset and a training pipeline for both agents. The coding agent is fine-tuned on high-quality text-CAD pairs, while the clarifying agent is trained using synthetic ambiguity trajectories that simulate clarification dialogues. This manuscript strives to analyze a broad topic related to reliable CAD generation from natural language and evaluates the proposed system using metrics such as Chamfer distance and invalidity ratio. Experimental results show that proactive clarification improves robustness to ambiguous prompts and produces more accurate CAD models compared with single-model baselines and strong frontier models.

**Compliance With Llm Reviewing Policy:**

Affirmed.

**Final Justification:**

Thanks for the response, I've increased my score.

**Key Questions For Authors:**

1.The clarification agent is trained and evaluated using LLM-simulated users. Could the authors provide results or analysis using real human interactions? Evidence that the system performs well with real users would significantly strengthen the empirical validity of the work.
2.The experiments focus on CadQuery as the CAD representation. How well would the proposed framework generalize to other CAD scripting languages or CAD systems? Demonstrating cross-framework applicability would increase the practical significance of the method.
3.How often does the clarifying agent ask unnecessary or redundant questions in practice? Additional analysis on interaction efficiency or user burden would help evaluate the usability of the system.
4.Could the authors provide more detailed analysis of failure cases where clarification does not resolve the ambiguity or where the final CAD program remains invalid? Understanding these limitations would help assess the robustness of the approach.

**Limitations:**

yes

**Strengths And Weaknesses:**

The paper is technically sound and proposes a well-motivated framework for improving text-to-CAD generation under ambiguous natural language specifications. The two-agent design, which separates clarification and code generation, is logically formulated and supported by experiments that evaluate both geometric fidelity and execution validity using established metrics such as Chamfer distance and invalidity ratio. The dataset construction pipeline also includes several quality control steps such as leakage checking and completeness verification, which improves the reliability of the training data. The presentation is generally clear, with a structured description of the pipeline and training procedure, and the experiments compare the proposed system with several strong baselines, including frontier models. The work addresses a practical and relevant problem in CAD automation, and the proactive clarification mechanism has potential practical value because real-world engineering descriptions are often incomplete. Although the core components build on existing LLM and agent-based paradigms, the integration of a clarification agent with CAD code generation and the accompanying dataset contribute a meaningful and potentially useful direction for text-to-CAD systems.
Despite the promising results, several limitations reduce the overall strength of the contribution. The evaluation relies heavily on synthetic ambiguous prompts and simulated user responses generated by large language models, which may not fully reflect real human interactions or real-world ambiguity patterns. This raises questions about the robustness of the clarification agent in realistic settings. In addition, the method mainly combines existing techniques such as supervised fine-tuning, agent-based prompting, and LLM-generated datasets, so the methodological novelty is somewhat limited and the contribution is primarily system-level rather than conceptual. The experimental section focuses on a specific dataset and CAD representation (CadQuery), which may limit the generalizability of the approach to other CAD frameworks or design environments. Finally, while the system shows strong performance improvements, the paper provides limited analysis of failure cases and does not deeply investigate when the clarification mechanism may introduce unnecessary interaction or errors.

---

> ### Author Rebuttal · Authors · 2026-03-31
>
> We sincerely thank Reviewer for the detailed feedback and recognize our paper as well-motivated with clear presentation and high-quality dataset pipeline
>
> **W1**: Thank you for pointing this out. To address this concern, we added **human-centered experiments** in which both the ambiguous prompts and the user responses are provided by real annotators rather than LLM simulators.
>
> Specifically, we sampled 100 examples from our high-quality Text-to-CAD dataset. For each example, we provided the example, including the original unambiguous prompt with the reference image to human annotators, since the original prompt contains the precise dimensions. We then asked multiple annotators to rewrite ambiguous prompts. The resulting clarification questions and corrected prompts were further evaluated by CAD experts. We also report the final Chamfer Distance after code generation.
>
> In this human evaluation, we compare **ProCAD-clarifier** against three baselines, while fixing the downstream generation model as **ProCAD-coder** for all methods. As shown in the tables below, our method continues to achieve the strongest performance. We will add these experiments and more analysis in the revision.
>
> | Model | Efficiency | Resolution | Mean CD  | Median CD  | Invalid Ratio  |
> | --- | --- | --- | --- | --- | --- |
> | **ProCAD-clarifier** | 0.760 | 0.787 | 0.00128 | 0.00009 | 12.2% |
> | Claude-sonnet-4-5 | 0.598 | 0.700 | 0.00972 | 0.00011 | 12.2% |
> | GPT-4o-mini | 0.251 | 0.287 | 0.01381 | 0.00014 | 19.5% |
> | Qwen2.5-7B | 0.016 | 0.000 | 0.01398 | 0.00369 | 14.6% |
>
> **W2**:  In the LLM-based text-to-CAD literature, most prior work in the related work section focuses on direct one-shot generation with standard supervised fine-tuning or RL with GRPO. In contrast, our paper is **one of the first** to explicitly formulate text-to-CAD under ambiguous specifications, and to introduce an agentic framework together with a finite-horizon Markov decision process formulation. The contribution is therefore not only a system design, but also a novel problem formulation for reliable CAD generation under ambiguity.
>
> In addition, the paper contributes a broader conceptual message for this research direction. As shown in Tables 1, 8 and 9, our results suggest that progress in text-to-CAD is not only about scaling the computation, but also about improving the quality of the text specification itself and explicitly resolving ambiguity before code generation. We view this as an shift for the field: from optimizing generation on noisy or weakly specified benchmarks to studying more precise and robust specification-aware generation.
>
> **W3**: Our new high-quality dataset is built through careful post-processing, leakage checking, and human evaluation. As shown in Tables 1, 8, and 9, current LLMs can learn effectively from substantially fewer samples on this high-quality, unambiguous dataset. To the best of our knowledge, both the unambiguous dataset and the synthetically ambiguous dataset are the first of their kind introduced in this paper.
>
> More importantly, the core idea of ProCAD is not specific to CadQuery, but applies to the broader problem of **clarification before executable CAD generation**. In our framework, the clarifying agent operates on the natural-language specification side, and is therefore largely agnostic to the underlying CAD representation.
>
> **W4**: We conducted a more detailed failure analysis. Across the test set, **12.3%** of examples receive either **0** or **0.5** under the **Resolution** metric. From these cases, we find:
>
> 1. **Most failures come from asking too few questions, not too many.**
> Among the failure cases, **99.4%** occur because the model asks **fewer questions than needed**, suggesting that unnecessary interaction is relatively rare.
> 2. **The main difficulty is multi-ambiguity detection.**
> The model performs well when there is a single ambiguity (**k=1: 98.2% success**) but is weaker when there are multiple ambiguities (**k=2: 77.1% success**).
> 3. **Some errors remain after successful clarification.**
> In a small number of cases, the model asks the correct question and receives the correct answer, but the final corrected specification is still wrong. A typical pattern is swapping axis values.
>
> For comparison, Qwen2.5-7B-Instruct almost never asks clarification questions at all. In the failure cases of Claude Sonnet 4.5, 16% involve unnecessary questions. Typical examples include asking about arc definitions, asking about units, or asking about extrusion direction when that aspect is already clear.
>
> Overall, these observations suggest that whether the clarifying agent asks too many or too few questions depends strongly on both the base model and the training setup. For our ProCAD-clarifier, the dominant failure mode is insufficient clarification, rather than excessive interaction.
>
> Please refer to our previous responses for the questions.

---

> > ### Author Rebuttal · Reviewer_cqV4 · 2026-04-05
> >
> > Thanks for the response, I've increased my score.

---

> > > ### Author Response · Authors · 2026-04-05
> > >
> > > We sincerely thank you for reviewing our work and for engaging with us during the discussion period. We are glad our response addressed your concerns, and your feedback helped improve the paper.

---

### Official Review · Reviewer_qCQM · 2026-03-12

**Soundness:** 3
**Presentation:** 3
**Significance:** 3
**Originality:** 2
**Overall Recommendation:** 4
**Confidence:** 4

**Summary:**

This paper proposes ProCAD, a two-agent framework for robust text-to-CAD generation under ambiguous natural-language specifications. The system separates the task into two components: (1) a clarifying agent that detects missing or conflicting geometric constraints and asks targeted questions, and (2) a coding agent that generates CadQuery programs from the corrected specification.

The authors also introduce a 10k high-quality Text-to-CadQuery dataset, constructed via a semi-automatic pipeline using VLM-generated descriptions and verification with Chamfer distance checks. The clarifier is trained using agentic supervised fine-tuning on clarification trajectories generated with a simulated user.

Experiments show that proactive clarification improves robustness to ambiguous prompts and significantly reduces invalid code generation. The full ProCAD system outperforms several strong baselines, including Claude Sonnet 4.5 and GPT-4o-mini, in terms of geometric fidelity and invalidity ratio.

**Compliance With Llm Reviewing Policy:**

Affirmed.

**Final Justification:**

added score by 1 for inclusion of human studies

**Key Questions For Authors:**

How well does the clarifying agent perform on real human prompts, rather than LLM-generated ambiguous prompts?

How sensitive is the system to different types of ambiguity beyond missing or conflicting dimensions?

Was any human expert validation conducted on the generated dataset?

What is the trade-off between clarification rounds and generation quality?

**Limitations:**

Evaluation relies on LLM-simulated user interactions rather than real user studies.

The ambiguity model covers only a limited subset of realistic prompt errors.

**Strengths And Weaknesses:**

## Strengths:

Well-motivated problem: The paper addresses ambiguity in natural-language CAD specifications, an important but underexplored issue in text-to-CAD systems.

Clear system design: The two-agent architecture (clarification → code generation) is sensible.

High-quality dataset pipeline: The dataset construction includes deduplication, leakage checks, and geometric verification, a good standard for CAD workflow.

## Weaknesses:

Reliance on simulated users: Ambiguous prompts and user responses are generated using LLM simulators rather than real human interactions.

LLM-based evaluation metrics: Interaction quality metrics rely on LLM-as-judge, which may introduce bias.

Limited ambiguity coverage: The experiments mainly consider two ambiguity types.

Limited real-world validation: The paper does not evaluate the system with human CAD users.

---

> ### Author Rebuttal · Authors · 2026-03-31
>
> We sincerely thank the Reviewer for the detailed feedback and recognize our paper as well-motivated, with a clear system design and a high-quality dataset.
>
> **W1**:  Thank you for pointing this out. To address this concern, we added **human-centered experiments** in which both the ambiguous prompts and the user responses are provided by real annotators rather than LLM simulators.
>
> Specifically, we sampled 100 examples from our high-quality Text-to-CAD dataset. For each example, we provided the example, including the original unambiguous prompt with the reference image to human annotators, since the original prompt contains the precise dimensions. We then asked multiple annotators to rewrite ambiguous prompts. The resulting clarification questions and corrected prompts were further evaluated by CAD experts.
>
> In this human evaluation, we compare **ProCAD-clarifier** against three baselines, while fixing the downstream generation model as **ProCAD-coder** for all methods. As shown in the tables below, our method continues to achieve the strongest performance. We will add these experiments and more analysis in the revision.
>
> | Model | Efficiency | Resolution | Mean CD  | Median CD  | Invalid Ratio  |
> | --- | --- | --- | --- | --- | --- |
> | **ProCAD-clarifier** | 0.760 | 0.787 | 0.00128 | 0.00009 | 12.2% |
> | Claude-sonnet-4-5 | 0.598 | 0.700 | 0.00972 | 0.00011 | 12.2% |
> | GPT-4o-mini | 0.251 | 0.287 | 0.01381 | 0.00014 | 19.5% |
> | Qwen2.5-7B | 0.016 | 0.000 | 0.01398 | 0.00369 | 14.6% |
>
> **W2**:  In the new human-centered experiments, the interaction quality metrics (**Efficiency** and **Resolution**) are evaluated by human experts, and the conclusions remain the same: our method consistently outperforms the baselines. In addition, the Resolution metric is closely related to the final geometric fidelity, so the improved Chamfer Distance provides an additional proxy showing that the clarified prompt better matches the target shape.
>
> We also note that our LLM-based evaluation is designed to reduce judging difficulty. For **Efficiency**, the judge is asked to compare the clarification questions against the ground-truth missing information, rather than making a fully open-ended quality judgment. For **Resolution**, we use a coarse three-level score {0, 0.5, 1}, which potentially further reduces sensitivity.
>
> Finally, we randomly sampled 100 test examples from the synthetic dataset and compared LLM-based evaluation against human evaluation of interaction quality. The two agree on more than **96%** of the samples, which provides additional evidence that the LLM-based metrics are reasonably reliable in our setting.
>
> **W3**:  We believe these two types capture a large portion of the ambiguity that is most critical in parametric CAD. In modern CAD systems like SolidWorks, sketches are commonly tracked as under-defined, fully defined, or over-defined states. There are certainly other forms of linguistic ambiguity. In this paper, our focus is on the ambiguity types that are most specific to CAD models, namely those directly tied to the dimensions and geometric constraints required to make a model fully defined. Although we group them into two high-level categories, these categories already include multiple concrete subtypes in our dataset. Across these categories, the key dimensions and constraints involve extrusion depth, length, radius, origin, and general coordinates.
>
> **W4**: As discussed in our response to Weakness 1, we have added a human-centered experiment in which the ambiguous prompts and responses are provided by real annotators, and our method remains stronger than the baselines in this setting. A more comprehensive study with real CAD users from diverse backgrounds is an important next step for this work, and we already discussed this in Lines 437–439.
>
> **Q1**: Please see the previous response.
>
> **Q2**: Please see our response to Weakness 3. It is also worth noting that even within our current setting, some samples already require nontrivial geometric reasoning rather than simple dimension matching. This further suggests that the two ambiguity classes studied here are not overly narrow in practice.
>
> **Q3**: Yes. We have human expert validation on the 10K high-quality dataset as shown in Figure 2. We note that we are one the first papers to involve human expert validation among recent text-to-cad papers.
>
> **Q4**: In our formulation, we assume that users provide correct answers as long as the clarification questions are clear. Under this assumption, once the missing constraints have been identified, additional clarification rounds do not add useful information and may instead increase context length and the risk of hallucination. Therefore, the goal is to ask the smallest sufficient set of targeted questions, rather than to prolong the interaction. Also empirically, in our experiments with Sonnet as the clarifier, we do not observe better generation quality from using more rounds.

---

> > ### Author Rebuttal · Reviewer_qCQM · 2026-04-03
> >
> > looks good. I added 1 points.
> >
> > I still think you should strive to get naturalistic interaction traces, rather than asking people to "mess up" a perfect instruction, why not just ask people to give instruction, then clarify as they would naturally ?

---

> > > ### Author Response · Authors · 2026-04-04
> > >
> > > We sincerely appreciate your positive acknowledgment and your decision to raise the score. We will include these additional details in the revised paper.
> > >
> > > We used the full instruction so that annotators could understand the **exact dimensions**, thereby ensuring a fixed ground-truth CAD model for objective evaluation. If annotators instead wrote instructions freely, the setup would be more naturalistic, but we would lose a well-defined reference CAD model, making Chamfer Distance less reliable and the evaluation of ProCAD-coder more subjective. We therefore chose this controlled human experiment as an intermediate step.
> > >
> > > We agree that collecting larger-scale naturalistic interaction traces is an important next step. One future direction may start from CAD drawings (which is also the starting point in CAD practice) and then ask annotators to write free-form textual descriptions. This would enable more naturalistic language data while still preserving a fixed target geometry for large-scale training and evaluation.
> > >
> > > Thank you again for the constructive feedback that helped strengthen our work!

---

### Decision · Program_Chairs · 2026-04-30

**Decision:**

Accept (regular)

**Comment:**

The paper introduces ProCAD, a proactive two-agent framework designed to handle ambiguous or under-specified prompts in text-to-CAD systems. Existing models reactively generate CAD models and often hallucinate missing dimensions or constraints when user instructions are incomplete. To mitigate this, ProCAD utilizes a proactive "clarifying agent" to ask targeted questions to resolve missing details and logical conflicts before a "drafting agent" generates the final CadQuery code.

The reviewers generally praised the practical motivation and the architectural design of the framework. A reviewer questioned the specific ranking mechanisms and the robustness of the interaction protocols used to evaluate the clarifying agent's effectiveness. In their rebuttal, the authors conducted a comprehensive new study during the rebuttal phase. This new study rigorously evaluated the interaction protocols and validated the claims regarding the effectiveness of the clarifying step.